# Simulation-driven design of stabilized SARS-CoV-2 spike S2 immunogens

Xandra Nuqui [1,8], Lorenzo Casalino [2,8], Ling Zhou [3], Mohamed Shehata[2], Albert Wang [4], Alexandra L. Tse [4], Anupam A. Ojha[1], Fiona L. Kearns[2], Mia A. Rosenfeld[1,5], Emily Happy Miller[4,6], Cory M. Acreman[3], Surl-Hee Ahn [7], Kartik Chandran[4], Jason S. McLellan [3] & Rommie E. Amaro [1,2] ✉

The full-length prefusion-stabilized SARS-CoV-2 spike (S) is the principal antigen of COVID-19 vaccines. Vaccine efficacy has been impacted by emerging variants of concern that accumulate most of the sequence modifications in the immunodominant S1 subunit. S2, in contrast, is the most evolutionarily conserved region of the spike and can elicit broadly neutralizing and protective antibodies. Yet, S2's usage as an alternative vaccine strategy is hampered by its general instability. Here, we use a simulation-driven approach to design S2-only immunogens stabilized in a closed prefusion conformation. Molecular simulations provide a mechanistic characterization of the S2 trimer's opening, informing the design of tryptophan substitutions that impart kinetic and thermodynamic stabilization. Structural characterization via cryo-EM shows the molecular basis of S2 stabilization in the closed prefusion conformation. Informed by molecular simulations and corroborated by experiments, we report an engineered S2 immunogen that exhibits increased protein expression, superior thermostability, and preserved immunogenicity against sarbecoviruses.

Coronaviruses (CoVs) are a group of enveloped viruses featuring a positive-strand RNA genome[1]. Several human CoVs (HCoVs) belonging to the α (229E and NL63) and β (OC43 and HKU1) genera endemically circulate every year, causing upper respiratory illness[2]. Over the first two decades of the 21st century, outbreaks of other β-CoVs, such as the Middle East respiratory syndrome (MERS)-CoV and the severe acute respiratory syndrome (SARS)-CoV viruses, led to severe cases of respiratory illnesses characterized by a high mortality rate[3]. More recently, COVID-19—the disease resulting from infection with another β-CoV, SARS-CoV-2—has caused at least 770 million cases and 6.9 million deaths worldwide since the virus was first identified[4]. Vaccines have provided protection against severe COVID-19 disease,

hospitalization, and mortality[5,6], but their efficacy has been challenged by the continual emergence of several SARS-CoV-2 variants of concern (VOCs), including Alpha, Beta, Delta, Gamma, multiple Omicron sublineages BA.1, BA.2, BA.3, BA.4, BA.5, and the currently dominant EG.5, FL.1.5.1, and XBB.1.16[7]. Therefore, while bivalent boosters may help recover some degree of efficacy, a more direct approach to designing pan-coronavirus vaccines is required for combating the relentless evolution of this pathogen[8].

The primary antigen used in most of the COVID-19 vaccines is an engineered version of the SARS-CoV-2 spike (S), a homotrimeric class I fusion glycoprotein stabilized in the prefusion state[9,10]. Prominently extending from the virus surface and responsible for cell entry[11], the

[1]Department of Chemistry and Biochemistry, University of California San Diego, La Jolla, CA, USA. [2]Department of Molecular Biology, University of California San Diego, La Jolla, CA, USA. [3]Department of Molecular Biosciences, The University of Texas at Austin, Austin, TX, USA. [4]Department of Microbiology and Immunology, Albert Einstein College of Medicine, Bronx, NY, USA. [5]Laboratory of Computational Biology, National Heart, Lung and Blood Institute, National Institutes of Health, Bethesda, MD, USA. [6]Department of Medicine, Division of Infectious Diseases, Albert Einstein College of Medicine, Bronx, NY, USA. [7]Department of Chemical Engineering, University of California Davis, Davis, CA, USA. [8]These authors contributed equally: Xandra Nuqui, Lorenzo Casalino. ✉e-mail: ramaro@ucsd.edu

spike serves as the principal target of the host's humoral immune response. The spike is composed of two subunits, S1 and S2. The former initiates viral entry by engaging host cell receptors, and the latter mediates fusion between viral and host membranes upon cleavage at the S1/S2 furin site, shedding of S1, further proteolytic cleavage at the S2′ site, and refolding from a prefusion to postfusion conformation[9–11]. On one hand, S1 is immunodominant as it encompasses the two domains eliciting the majority of anti-CoV neutralizing antibodies[12–15], the N-terminal domain (NTD) and the receptor-binding domain (RBD). On the other hand, due to selective immune pressure[16,17], most of the mutations across different CoV strains and SARS-CoV-2 VOCs occur in S1[2,18–20]. As a result, the acquired humoral immunity gained from S1-directed antibodies is often closely related to the specific CoV strain or SARS-CoV-2 VOC responsible for the infection. Conversely, the S2 subunit is highly conserved and less prone to accumulate sequence modifications, offering the opportunity to design S2-based immunogens that provide broader protection against different SARS-CoV-2 VOCs and, possibly, CoV strains[21–24]. Several studies have characterized cross-reactive antibodies targeting the S2 subunit's stem helix[18,21,23,25], fusion peptide[24], and the usually occluded 'hinge' region at the trimer apex spanning residues 980–1006[26–28].

Although immunogenically appealing, prefusion S2 is metastable as it irreversibly transitions to a stable, postfusion conformation when S1 is removed[29]. The S2 trimer adopts an elongated baseball-bat-like conformation, where the heptad repeat 1 (HR1) and central helices (CHs) refold at the hinge region, transitioning from a bent hairpin into a continuous, extended coiled-coil helical bundle[18,29]. This structure is framed by the HR2 motif at the helix-helix interfaces, yielding a rigid six-helix bundle in the regions where HR2 maintains its helical fold[18,29]. Introduction of two consecutive prolines within the HR1/CH hinge point (K986P and V987P) in the S2 subunit of the engineered SARS-CoV-2 'S-2P' spike proved a successful strategy to prevent progression to the postfusion state[9,30]. Further stabilization of the prefusion conformation, along with improved cellular expression, has more recently been accomplished upon the incorporation of four additional prolines in the S2 subunit (F817P, A892P, A899P, A942P), resulting in the SARS-CoV-2 'HexaPro' spike[31]. In this construct, the three S1 subunits sit atop and seal together the HexaPro S2 subunits nestled underneath. Interestingly, previous HDX-MS experiments showed that the SARS-CoV-2 HexaPro spike can also sample an open conformation where the S2 trimer interfaces become solvent accessible[28]. This may suggest that, in the absence of S1, the HexaPro S2-only trimer would most likely dissociate into individual monomers. To stabilize the HexaPro S2-only trimer in the prefusion conformation, McLellan and coworkers recently introduced an interprotomer disulfide bond via two cysteine substitutions (S704C and K790C) and an interprotomer salt bridge with R765 upon Q957E substitution (Fig. 1)[32]. However, the apex of the resulting construct (HexaPro stabilized stem without the stalk), termed 'HexaPro-SS-Δstalk', was shown to splay apart, adopting semi-open and open conformations[32]. The pronounced conformational plasticity exhibited by the HexaPro-SS-Δstalk S2 prefusion trimer underscores the potential for improving its stability by inducing a shift in the equilibrium towards a closed conformation.

Opening of the trimer, or 'breathing,' is a motion observed in several class I fusion glycoproteins, including influenza hemagglutinin (HA)[33,34], HIV-1 Env[35], respiratory syncytial virus (RSV) F[36], as well as the full-length SARS-CoV-2 spike[26–28]. Silva et al.[26] showed that accessibility of the full-length SARS-CoV-2 spike's hinge epitope necessitates both RBD transitioning to the 'up' conformation and S2 subunits 'breathing.' This aligns with HDX-MS experiments showing the existence of an open conformation of the full-length spike where the S2 apex is solvent-exposed[28]. However, S2 apex epitopes elicit non-neutralizing antibodies[26,27], which are dominant in the B cell population[27,37] and can interfere with the binding of neutralizing antibodies[38,39]. This is also the case for other trimer interface epitopes in influenza HA[34] and human metapneumovirus (hMPV) F[37]. Hence, preventing the S2 trimer's apex opening may offer the foundations for the design of improved S2-based immunogens that may also afford binding and elicitation of quaternary epitope-specific antibodies.

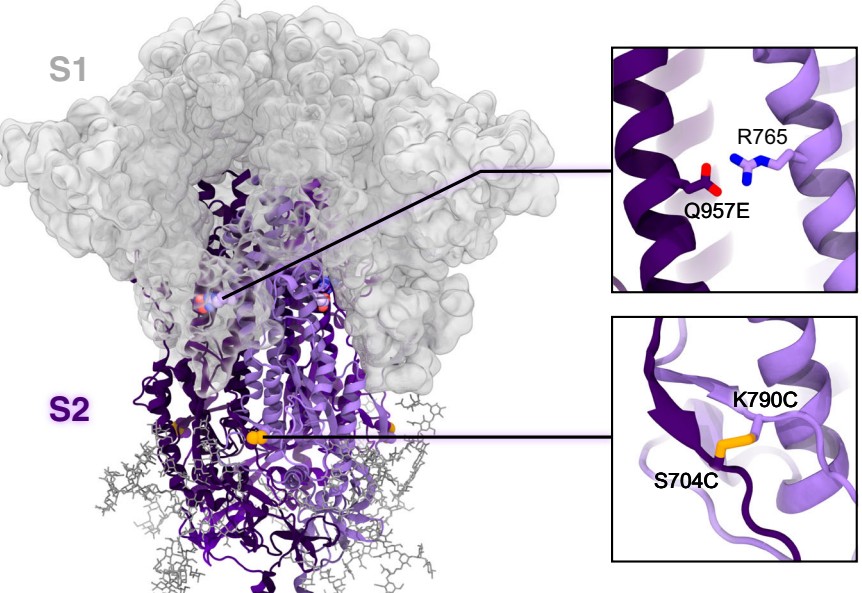

**Fig. 1 | HexaPro-SS-Δstalk S2-only trimer and context within the SARS-CoV-2 spike protein.** The model of the HexaPro-SS-Δstalk S2-only trimer in the closed prefusion conformation is depicted with a cartoon representation where the protomers are highlighted with different shades of purple. We note that the spike's S1 subunit, here shown with a gray transparent surface only to provide context, is not present in this construct. N-glycans linked to S2 are shown as dark gray sticks. Glycans linked to S1 are omitted for clarity. HexaPro-SS-Δstalk is an S2-only construct derived from the prefusion-stabilized SARS-CoV-2 HexaPro spike[32]. The distinctive substitutions (i.e., S704C, K790C, Q957E) incorporated in the HexaPro-SS-Δstalk[32] are highlighted in the panels on the right. They stabilize S2 in a prefusion trimer assembly, preventing protomer dissociation via the formation of interprotomer salt bridges and disulfide bonds.

Here, we use a simulation-driven approach to characterize the opening mechanism of the SARS-CoV-2 S2 trimer and design an S2-based immunogen that is stable in the closed prefusion conformation. Weighted ensemble[40] (WE) molecular dynamics (MD) simulations allowed us to characterize how the S2 trimer apex splays open along an asymmetric pathway. We use this dynamical information as a basis to design two cavity-filling tryptophan substitutions (V991W and T998W) that alter the conformational landscape of S2, stabilizing the closed prefusion conformation. These substitutions also provide enhanced thermostability and cellular expression, allowing complete structural characterization via cryo-EM. Subsequent neutralization assays showed that sera from mice immunized with the newly designed S2 antigen significantly neutralizes recombinant vesicular stomatitis viruses (rVSVs) bearing spikes from SARS-CoV-2 Wuhan-1 and Omicron BA.1. Overall, by integrating cutting-edge computational methodologies and a comprehensive array of experiments, our work offers a thorough understanding of the mechanistic, structural, dynamical, and immunogenic aspects of aptly engineered S2-only antigens. The findings presented here open new avenues for devising improved S2-based immunogens, thereby holding promise for the development of pan-coronavirus vaccines.

## Results

### Prefusion-stabilized S2 asymmetrically transitions to a splayed open conformation

To probe the stability of the prefusion-stabilized HexaPro-SS-Δstalk S2-only base construct, we first sought to characterize the conformational plasticity of the closed conformation within the nanosecond timescale by performing all-atom conventional MD simulations. A model of HexaPro-SS-Δstalk in the closed conformation was built from the S2 domain of the cryo-EM structure of the SARS-CoV-2 HexaPro spike (PDB ID: 6XKL[31]) integrated with the fusion peptide's residue coordinates grafted from the SARS-CoV-2 3Q-2P spike cryo-EM structure (PDB ID: 7JJI[41]). The distinctive substitutions engineered in the HexaPro-SS-Δstalk[32] (i.e., S704C, K790C, Q957E) were then incorporated using Visual Molecular Dynamics (VMD)[42], and S2's N-linked glycans were added as described by Casalino et al.[11]. The final system (Fig. 1), comprising ~274,000 atoms with explicit water molecules and ions, was subjected to four, ~600 ns-long MD simulation replicates (Supplementary Table 1). In these simulations, we observed one protomer's apex transiently swinging away from the other two protomers (Supplementary Fig. 1); however, the trimer did not adopt an open conformation where the three protomers are splayed apart. Notably, the slight detachment of one protomer from the others was observed only once throughout a combined sampling of 2.6 μs, suggesting that trimer opening is a rare (or infrequent) event when investigated by means of conventional MD. Thus, to elucidate the complete transition of HexaPro-SS-Δstalk from a closed conformation to an open conformation, an enhanced sampling method was necessary.

Exploration of a system's conformational space, such as the one conducted with conventional MD, is limited by high free energy barriers that separate distinct minima in which a system may remain trapped. As large conformational transitions occurring in the millisecond-to-second timescale are computationally prohibitive via conventional MD, enhanced sampling methods may be employed instead[40,43,44]. Therefore, we first attempted to inquire into the conformational transition of HexaPro-SS-Δstalk from the closed to the open conformation using Gaussian accelerated MD (GaMD). GaMD is a method that applies a harmonic boost potential to smooth energy barriers on the potential energy surface, expanding the accessibility of long-timescale dynamic motions[44]. However, we found that the applied boost potential at the highest acceleration level was insufficient to induce sampling of open conformations (Supplementary Fig. 1 and Supplementary Tables 1, 2). Hence, supported by our previous success in studying the SARS-CoV-2 spike's RBD down-to-up

transition[45], we opted to investigate the opening of the HexaPro-SS-Δstalk S2 trimer using the WE path-sampling method[40,46]. By running many short (0.1 ns) stochastic MD simulations in parallel, and iteratively replicating trajectories that have explored new regions of configurational space defined by appropriately binned progress coordinates, the WE strategy facilitates atomically detailed sampling of large conformational changes without alteration of the system's free energy[47].

To enhance sampling of open conformations of the initially closed HexaPro-SS-Δstalk construct, we defined two progress coordinates (Fig. 2a, b): (1) the area of the triangle formed by the Cα atoms of the three P987 residues (one for each protomer) residing at the apex of the CHs, and (2) the RMSD of the CHs with respect to the CHs as in the open conformation resolved in the crystal structure (PDB ID: 8U1G[32]). WE simulations of HexaPro-SS-Δstalk were performed with WESTPA 2.0[48,49] and Amber20[50], generating 180 continuous opening pathways across 45 μs of aggregate sampling time (for a complete description of WE simulations, see "Extended Computational Methods" in the Supplementary information, Supplementary Fig. 2 and Supplementary Tables 1, 3).

The wealth of atomic-level information provided by these simulations allowed us to characterize the mechanism by which the HexaPro-SS-Δstalk S2-only immunogen adopts a splayed open conformation (Supplementary Movie 1, Supplementary Data 1). Starting from a closed conformation (Fig. 2c), the initial detachment of chain-A leads to an asymmetric, partially open conformation (Fig. 2d) until chain-B and chain-C separate to form the open conformation (Fig. 2e, Supplementary Movie 1). Notably, asymmetric breathing was also recently reported for influenza HA[33]. In the open conformation, the distances between the P987 residues increase from values close to ~20 Å (as in the closed conformation) to values in the range of ~35 to ~40 Å. Remarkably, the open conformation obtained through WE simulations closely aligns with the open conformation resolved using X-ray crystallography (Supplementary Fig. 3)[32]. Interestingly, we also observed further open conformations where at least one of the trimer's protomers kept moving further away (Supplementary Fig. 4). The opening pathways were used for analyzing the contacts at the interfacial helices in the closed, partially open, and open conformations (Fig. 3a–c and Supplementary Figs. 5–7). Using PyContact[51], we examined interprotomer residue–residue interactions involving the CHs (residues 978–1030), upper helices (UHs, residues 738–783), and HR1 helices (residues 945–977). S2 trimer opening occurs upon a progressive breakage of interprotomer residue–residue interactions within the CHs from apex to base, following a trend from hydrophilic to hydrophobic (Fig. 3c). A salt bridge between R995 and D994 breaks before the hydrogen bond network consisting of Q1002, Q1005, T1006, and T1009 is broken until the loss of the hydrophobic interactions between L1012 and I1013 leads to the separation of chain-A. As expected, the helices in chain-A did not form strong interactions in the partially open conformations, and interactions throughout the trimer were almost null in the open conformation (Fig. 3c and Supplementary Figs. 5–7). The partially open conformations are characterized by CH_Chain-B forming an extensive interaction interface with the polar residues along UH_Chain-C (Fig. 3c and Supplementary Figs. 5–7). Interestingly, residues V991 and T998 in the CHs did not contribute to stabilizing interprotomer contacts in either the closed or partially open conformations (Fig. 3a–c). On the contrary, residues D994, R995, and Q1002, which are one or two helical turns away from V991 and T998, form stabilizing interprotomer interactions in both the closed and partially open conformations (Fig. 3a–c).

The opening mechanism of HexaPro-SS-Δstalk underscored two residues located in the proximity of the CH apex, V991 and T998, as best candidates for cavity-filling substitutions aimed at improving the stability of the closed conformation. Cavity-filling is a common strategy utilized to stabilize proteins by introducing hydrophobic

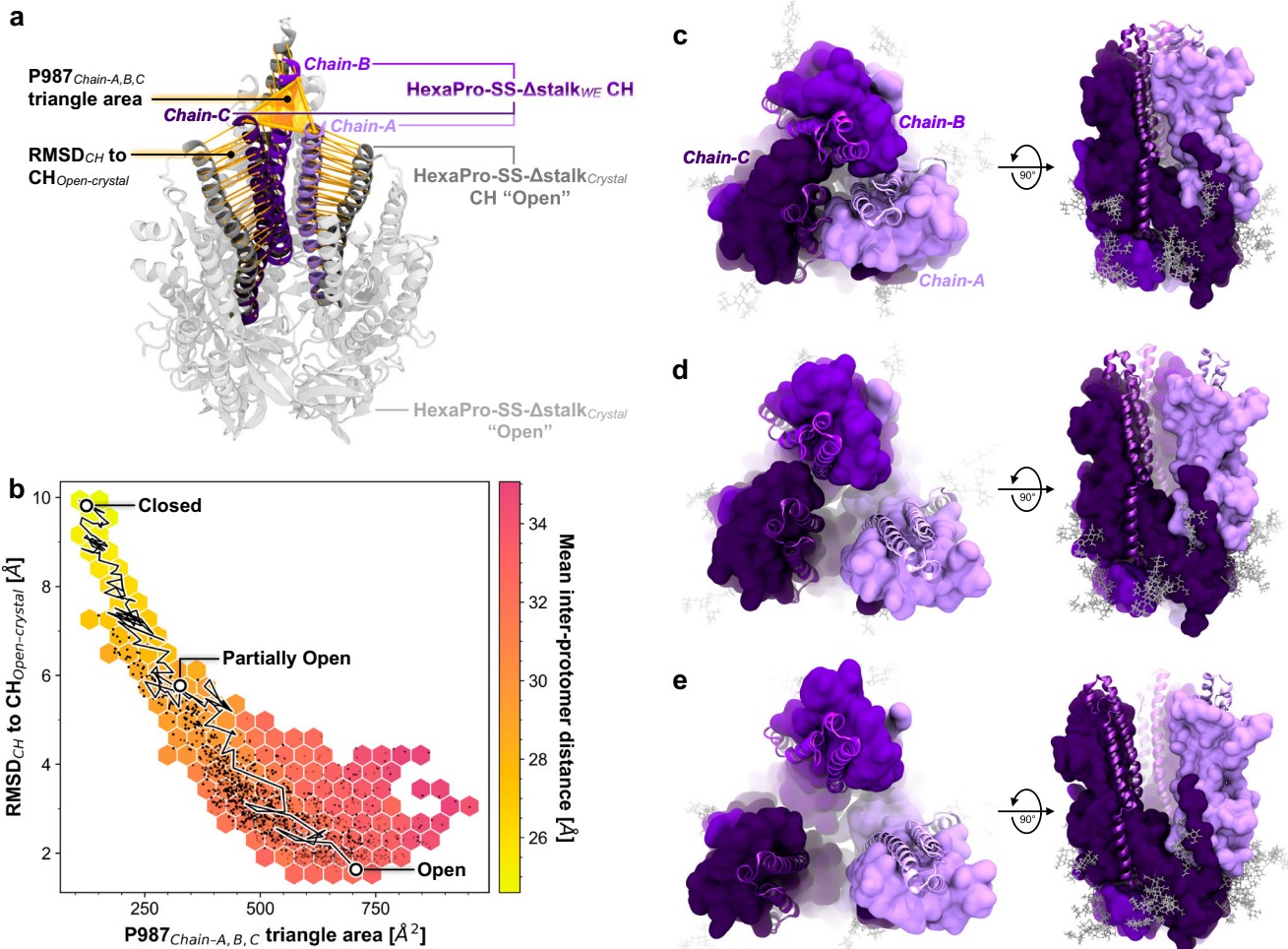

**Fig. 2 | HexaPro-SS-Δstalk S2 trimer opening. a** Progress coordinates used in the WE simulation of HexaPro-SS-Δstalk, namely P987$_{C\alpha}$ triangle area and RMSD$_{CH}$, are highlighted with yellow shapes (triangle and cylinders, respectively) drawn on top of the molecular structures. The CHs as in the closed conformation of the simulated construct are shown with shades of purple and are overlaid on the open conformation as in the crystal structure (colored in gray, PDB ID: 8U1G[32]). **b** Distribution of the conformations sampled in the opening pathways obtained from the WE simulation. Each black point represents a conformation sampled along an opening pathway. The RMSD of the CH to CH$_{Open-crystal}$ ($y$-axis) is plotted against

the area of the triangle formed by the P987$_{C\alpha}$ at the CH apex ($x$-axis). The hexagonal bin color is scaled to the mean interprotomer distance of the black data points within the respective bin. A trace of one of the successful pathways is shown as a black line with white points corresponding to the closed, partially open, and open conformations. **c**–**e** Molecular representation of closed (**c**), partially open (**d**), and open (**e**) conformations as highlighted in (**b**). Chain-A is depicted in light purple, chain-B in purple, and chain-C in dark purple. Residues 900–1030 encompassing HR1 and CH are illustrated with a cartoon representation, whereas the rest of the chain is shown as an opaque surface. Glycans are shown as gray sticks.

---

sidechains to occupy hollow regions within the protein[52,53]. This approach has been previously applied to other viral glycoproteins such as RSV F, HIV-1 Env[53,54], and SARS-CoV-2 S1[55]. Intrigued by this observation, we hypothesized that introducing a bulky hydrophobic sidechain at positions 991 and 998, such as in tryptophan, may contribute to forming novel, interprotomer, stabilizing interactions, which, in turn, could slow down opening or fully stabilize the S2 trimer in the closed prefusion conformation.

### Tryptophan cavity-filling substitutions slow down prefusion-stabilized S2 trimer opening

Aiming to stabilize the S2 trimer in the closed prefusion conformation, we designed three HexaPro-SS-Δstalk tryptophan variants. Three engineered models in the closed conformation were generated from the base construct: (1) *HexaPro-SS-V991W*, incorporating V991W substitution; (2) *HexaPro-SS-T998W*, including T998W substitution; (3) a double substitution variant, named *HexaPro-SS-2W*, containing both V991W and T998W substitutions. In a similar way to the base construct, we performed WE simulations of the three HexaPro-SS-Δstalk tryptophan variants to interrogate their stability and, possibly, opening

mechanism. Interestingly, each engineered S2 trimer swung away from the initial closed conformation to sample partially open and open conformations. We collected 149, 187 and 195 opening pathways across 35, 47, and 56 μs of aggregate sampling for HexaPro-SS-V991W, HexaPro-SS-T998W, and HexaPro-SS-2W, respectively (Supplementary Figs. 8–12 and Supplementary Tables 1, 3). The closed-to-open transition is consistent across all four simulated S2 constructs, with chain-chain separation proceeding stepwise. Although opening occurred, it is important to note that WE is a path-sampling approach used to enhance the sampling efficiency of conformational transitions that are rarely observed by means of conventional MD[40]. That is, sampling an open conformation does not itself imply that the closed conformation is unstable; rather, the WE method allowed us to observe a slow dynamic behavior that is physically accessible to the protein.

Our WE simulations indicate that trimer opening is somehow affected by the introduction of the V991W and T998W substitutions. There are differences in the aggregate sampling time to observe the first opening event during WE simulations (i.e., including the time spent in the closed state) (Supplementary Fig. 13a), with HexaPro-SS-T998W and HexaPro-SS-2W requiring an additional ~6.3 μs and ~4.9 μs

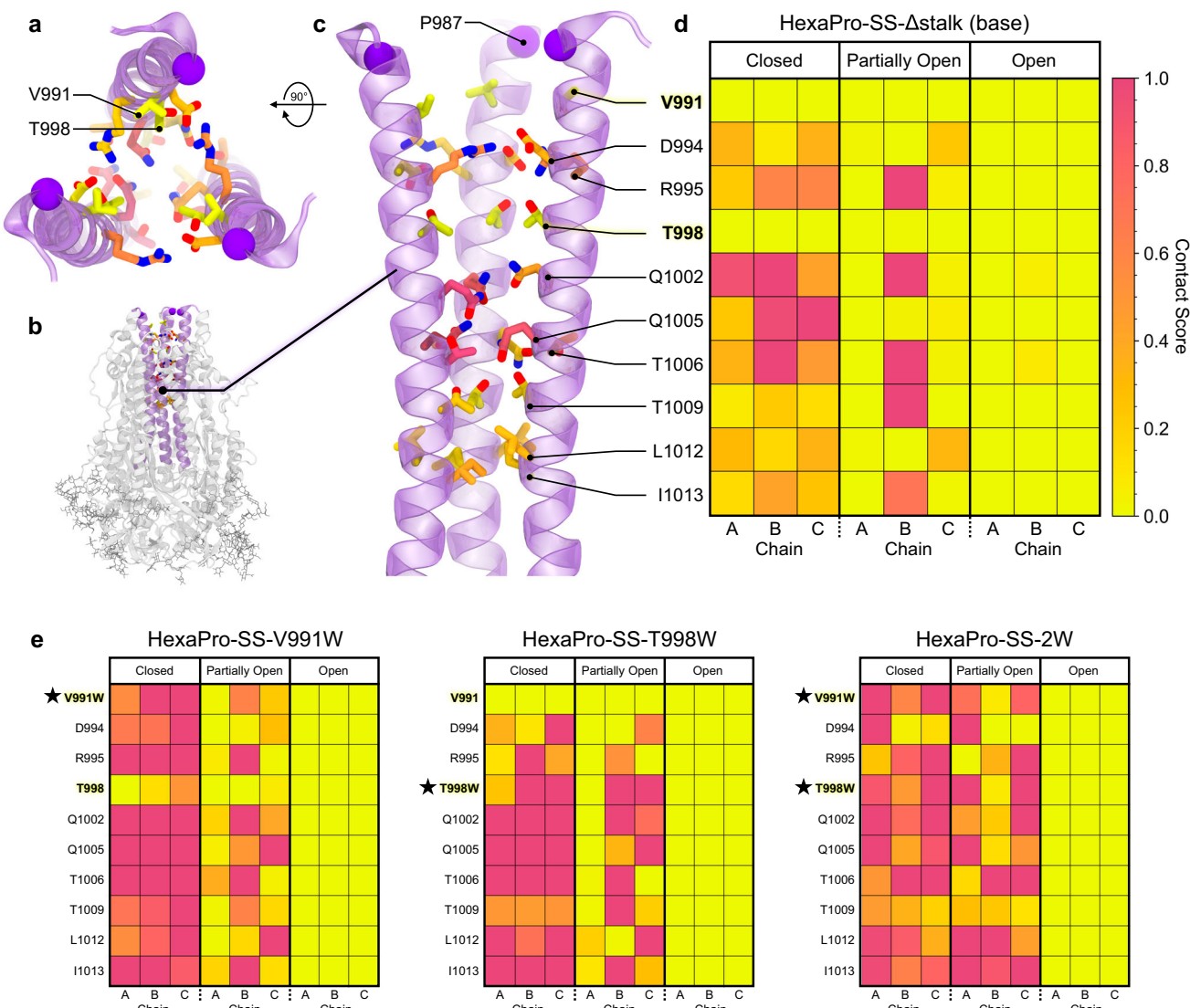

**Fig. 3 | Contacts between residues in the central helices during S2 opening.** **a**–**c** HexaPro-SS-Δstalk CHs in the closed conformation from a top-down view (**a**), side view in the context of the S2 trimer (gray cartoons) (**b**), and side view (**c**). CHs are highlighted with purple cartoons. The sidechains of the residue pointing toward the interior of the CHs are shown as sticks with the C atoms colored by the corresponding residue contact score as in the closed conformation of the HexaPro-SS-Δstalk. **d**, **e** Contact score heatmaps for the HexaPro-SS-Δstalk base construct (**d**) and HexaPro-SS-Δstalk tryptophan substitutions (**e**) in the closed, partially open, and open states. The scale ranges from 0 (weak or transient contacts) to 1 (persistent or extensive contacts, corresponding to the 95th percentile value of the HexaPro-SS-Δstalk contact scores). The star indicates the substituted residue.

of sampling, respectively, and a corresponding shift toward overall longer molecular times to opening (Supplementary Fig. 13b). This may also suggest more time spent in the closed state. In contrast, HexaPro-SS-V991W necessitates less sampling to open for the first time, approximately ~1.5 μs less than the base construct. Analysis of opening event durations as a measure of the molecular time required by the system to transition from the closed state to the open state (i.e., excluding the time spent within the closed state), shows a distribution slightly shifted toward longer durations for the three tryptophan variants (Supplementary Fig. 14), with HexaPro-SS-T998W exhibiting the longer mean duration followed by HexaPro-SS-2W (Supplementary Table 3). In this regard, no significant differences are observed for HexaPro-SS-V991W with respect to the base construct, suggesting that V991W might not affect the opening kinetics. One advantage of WE simulations is the possibility of estimating kinetics quantities such as rate constants underlying the investigated transition[48,56–58]. In our case, by iteratively reweighting the WE trajectories using the WE Equilibrium Dynamics (WEED)[56,58] protocol within WESTPA 2.0[48] to facilitate the convergence of WE simulations toward near equilibrium steady-state conditions[56,57], we were able to estimate the rate constant of the opening transition ($k_{opening}$) for each simulated system (see "Extended Computational Methods" in the Supplementary information for a detailed description, Supplementary Fig. 15 and Supplementary Table 3). We estimated that the opening of the HexaPro-SS-Δstalk S2 trimer occurs on the order of fractions of milliseconds [$k_{opening} = (4.5 \pm 1.5) \times 10^4\,\mathrm{s}^{-1}$]. Although the rate constant does not significantly change for HexaPro-SS-V991W [$k_{opening} = (1.1 \pm 0.5) \times 10^5\,\mathrm{s}^{-1}$], it becomes smaller for HexaPro-SS-2W [$k_{opening} = (7.0 \pm 2.3) \times 10^3\,\mathrm{s}^{-1}$] and HexaPro-SS-T998W [$k_{opening} = (3.4 \pm 0.1) \times 10^3\,\mathrm{s}^{-1}$], indicating that T998W induces a roughly 10-fold slower trimer opening. V991W alone does not appear to affect opening kinetics. Although we acknowledge that these estimations might be affected by sampling limitations, the calculated values align well with each other and single-molecule fluorescence resonance energy transfer (smFRET) studies reporting similar domain movements of class I fusion glycoproteins occurring in the millisecond-to-second time scales[59–61].

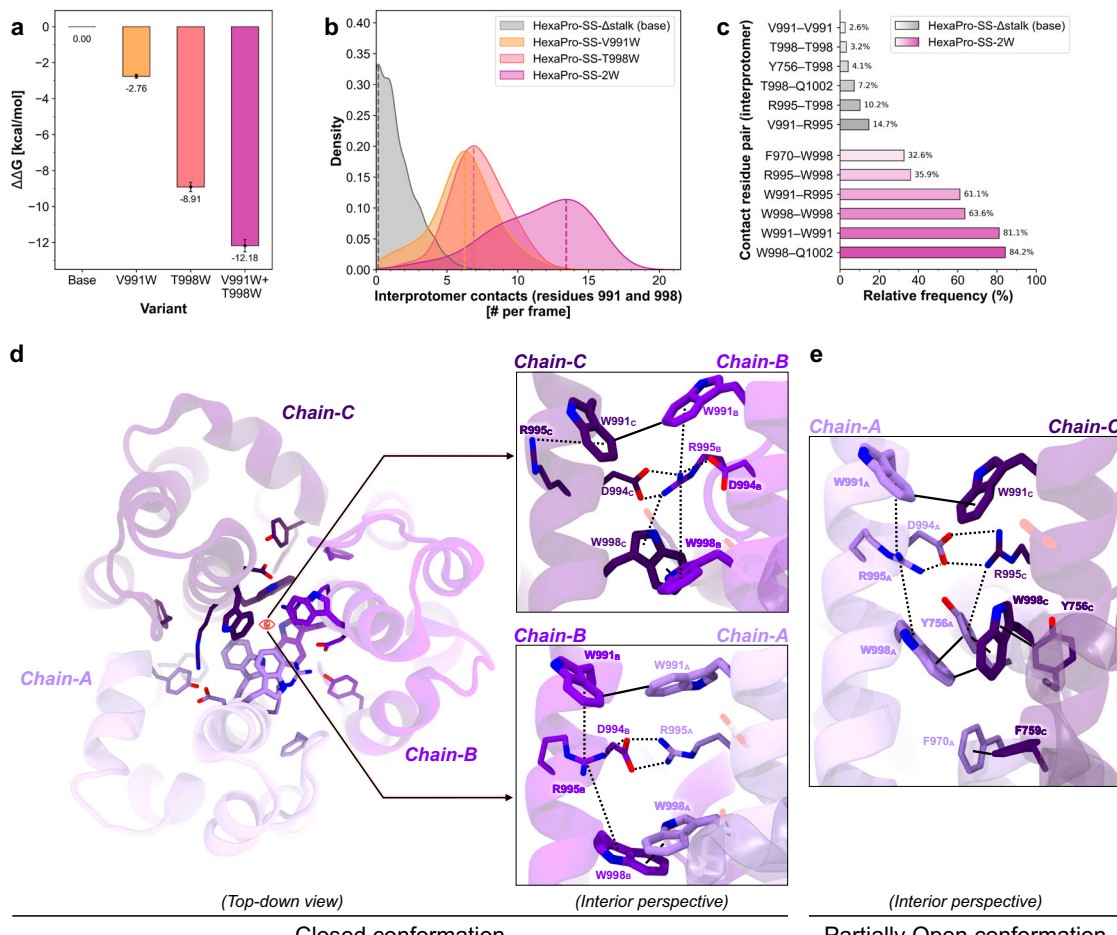

**Fig. 4 | V991W and T998W stabilize HexaPro-SS-Δstalk in the closed conformation. a** Thermodynamic stabilization of the closed prefusion conformation as imparted by each tryptophan variant relative to the base system (HexaPro-SS-Δstalk). For each variant, the value of $\Delta\Delta G_{mutation\text{-}folding}$ relative to the base construct is plotted with a colored bar indicating the free energy difference estimate (in kcal/mol) ± standard error as calculated from 1000 (forward and backward) non-equilibrium alchemical simulations. The standard errors, denoted by thin black lines, were calculated via bootstrapping with 100 drawn samples. **b** Distribution of the number of interprotomer contacts established per frame by residues at positions 991 + 998 from the ensemble of closed conformations extracted from respective WE simulations. Distributions are shown as kernel densities. **c** Relative frequency of occurrence (%) of the top six residue–residue interprotomer contacts

occurring in the closed conformations observed in HexaPro-SS-Δstalk and HexaPro-SS-2W WE simulations. Relative frequencies are calculated with respect to the total number of closed conformations extracted from the respective WE simulation. **d**, **e** Molecular representation of the most important interprotomer contacts in HexaPro-SS-2W in (**d**) the closed conformation and (**e**) the partially open conformation. Chain-A is depicted in light purple, chain-B in purple, chain-C in dark purple, and interacting residues are highlighted with sticks. Solid lines connect residues that form hydrophobic (π–π) interactions, whereas dashed lines indicate electrostatic (cation–π and salt bridge) interactions. A representative closed conformation of HexaPro-SS-2W was selected for this purpose upon clustering of closed conformations retrieved from the respective WE simulation (Supplementary Fig. 18).

As previously performed for the base construct (Fig. 3d), we looked at the interprotomer residue–residue interactions for the tryptophan variants to gain a mechanistic understanding of the impact of V991W and T998W on the opening pathways. As highlighted from the contact heatmaps in Fig. 3e and Supplementary Figs. 5–7, both V991W and T998W substitutions boost interprotomer interactions in the closed and partially open conformations with respect to the base construct. These interactions promote engagement at the respective substituted positions and the nearby residues located a few helical turns away (Fig. 3e). The increased interhelix interplay observed in the partially open conformations may underlie the longer opening durations observed in HexaPro-SS-T998W and HexaPro-SS-2W.

**V991W and T998W substitutions stabilize S2 in a closed prefusion conformation**

Beyond characterizing the trimer opening kinetics for the tryptophan variants, we sought to investigate whether V991W and T998W substitutions energetically stabilize HexaPro-SS-Δstalk in the closed

prefusion conformation. Computationally, this can be probed through non-equilibrium 'alchemical' free energy calculations where an amino acid (i.e., V or T) is gradually morphed into another one (i.e., W) in a set of non-equilibrium MD simulations, both in the context of the folded and unfolded protein (see "Extended Computational Methods" in the Supplementary information for a complete description)[62]. Relying on the notion that the free energy of a system is a state variable, it is possible to calculate the difference in the free energy of folding ($\Delta\Delta G_{mutation\text{-}folding}$) between the investigated variant and the base construct (closed prefusion HexaPro-SS-Δstalk), as illustrated in the thermodynamic cycle in Supplementary Fig. 16. Here, starting from an equilibrated closed conformation of HexaPro-SS-Δstalk, we performed non-equilibrium alchemical free energy calculations to predict the contribution of V991W and/or T998W substitutions to the folding free energy in the three engineered tryptophan variants in the closed conformation (Supplementary Fig. 17). As a result, in all three cases, the introduced substitutions are stabilizing (Fig. 4a). In ascending order (i.e., from the least to the most stabilizing), calculated $\Delta\Delta G_{mutation\text{-}folding}$

for HexaPro-SS-V991W, HexaPro-SS-T998W, and HexaPro-SS-2W, are −2.76 ± 0.11, −8.91 ± 0.26, and −12.18 ± 0.34 kcal/mol (Fig. 4a and Supplementary Table 3). Hence, these independent simulations predicted the double substitution variant, HexaPro-SS-2W, to be the most stable in the closed prefusion conformation.

For each system, we extracted all the closed conformations that were sampled during the respective WE simulation, including those that did not result in an opening event. Using this multi-microsecond-long conformational ensemble of closed conformations, we then analyzed interprotomer contacts involving CHs, UHs, and HR1 helices to decipher, at the molecular level, the stabilizing effect imparted by the tryptophan substitutions (Supplementary Figs. 19–21). Introducing V991W and T998W into HexaPro-SS-Δstalk directly contributes to stabilizing contacts, enhancing electrostatic and hydrophobic interaction networks (Fig. 4b–e, Supplementary Figs. 19–21 and Supplementary Movie 2). The tryptophan sidechains at positions 991 and 998 self-associate to form π–π hydrophobic clusters at the S2 apex surrounding R995 (Fig. 4c–e and Supplementary Movie 2). Notably, alongside forming a salt bridge with D994 from the adjacent protomer, the sidechain of R995 can also adopt an inward-oriented conformation, where it engages in a π–cation–π stack with W991 and W998 from both the same and neighboring protomers (Fig. 4d, e, Supplementary Figs. 19, 20 and Supplementary Movie 2). W998 is sometimes observed to swivel slightly outward, interacting with Y756 and F759 of the same chain and F970 from the adjacent chain. Interestingly, an extensive network of interactions is also preserved in the partially open conformations obtained from the WE simulation, where the presence of intra-protomer π–π and cation–π interactions modulate the orientation of the tryptophans' sidechain (Fig. 4e). Most importantly, the extensive network of π–π and cation–π interactions at the S2 apex is absent in the HexaPro-SS-Δstalk base construct (Fig. 4c and Supplementary Figs. 19, 20).

To experimentally probe the computational predictions, all three tryptophan variants were assessed relative to HexaPro-SS-Δstalk for expression yield in FreeStyle 293F cells (Fig. 5a, b) and thermostability through differential scanning fluorimetry (DSF) (Fig. 5c). In line with the stabilization trend predicted with free energy calculations, expression of HexaPro-SS-2W and, to a lesser extent, HexaPro-SS-T998W, led to higher yields than the base construct. In addition, the slightly right-shifted peaks of their size-exclusion chromatography (SEC) profiles hint at more compact structures (Fig. 5b). V991W alone does not significantly improve the expression yield of HexaPro-SS-V991W relative to the base construct. A similar trend is observed from the thermostability profile (Fig. 5c), where HexaPro-SS-2W exhibits the highest melting temperature relative to the control systems (+5 °C shift relative to HexaPro-SS-Δstalk and +16 °C shift relative to HexaPro), followed by HexaPro-SS-T998W, and HexaPro-SS-V991W.

Based on the SEC data that indicated a ~2-fold increase in purified protein, we attempted to characterize the 3D structures of both HexaPro-SS-T998W (Δstalk) and HexaPro-SS-2W (Δstalk) using cryo-EM. For the former, we did not obtain a high-resolution cryo-EM structure; indeed, the low-resolution 3D ab initio map shows twisted S2 single protomers (data not shown). On the other hand, for HexaPro-SS-2W, we successfully generated a high-resolution (2.8 Å) cryo-EM map from 1622 exposures collected on a 200 kV Glacios. The cryo-EM map of the HexaPro-SS-2W was processed with C3 symmetry and DeepEMhancer[63] sharpening. 2D classes, especially the top-down views, indicate that the closed prefusion conformation is predominant in HexaPro-SS-2W (Supplementary Fig. 22). This result is consistent with free energy calculations and DSF experiments. The structure highlights key residue interactions also observed in the WE simulation, showing a densely packed cluster of hydrophobic residues at the S2 trimer's apex (Fig. 5e–g). In this region, an extended network of π–π interactions encompassing W991, W998, Y756, F759, and F970 surrounds an ordered salt bridge (D994–R995) (Fig. 5e and

Supplementary Fig. 23). Examining the cryo-EM density of each mutation, V991W at the very apex of the S2 trimer shows clear cation–π interactions with R995 (Fig. 5f). T998W shows clear cryo-EM density of the sidechain, filling the cavity in between the CHs of the S2 trimer (Fig. 5g). By stabilizing the S2 trimer's apex with V991W and T998W, we were also able to resolve the fusion peptide proximal region (FPPR, residues 835–857). This region had been challenging to characterize in the full-length spike or HexaPro-SS due to its inherent flexibility.

## Sera of mice immunized with closed prefusion-stabilized S2-only antigen neutralize rVSV-SARS-CoV-2 variants

The impact of tryptophan substitutions on S2-only antigen immunogenicity was investigated. We immunized mice with the HexaPro-SS and HexaPro-SS-2W antigens (including the stalk region) and assayed the sera for neutralization against recombinant vesicular stomatitis viruses (rVSVs) bearing the coronavirus spikes of either SARS-CoV-2 Wuhan-1 or Omicron BA.1 variant (Fig. 6 and Supplementary Fig. 24). Against rVSV-SARS-CoV-2 Wuhan-1, both sera immunized with HexaPro-SS and HexaPro-SS-2W show a significant increase in neutralization (Fig. 6b) as compared to PBS-immunized sera, but do not differ with respect to each other. In contrast, only mice immunized with HexaPro-SS-2W demonstrated a significant increase in the neutralization of rVSV-SARS-CoV-2 Omicron BA.1 compared to PBS-immunized sera.

To evaluate the impact of the tryptophan substitutions on the preservation of the epitopes on the spike S2 subunit, we performed biolayer interferometry (BLI) experiments. We tested the binding of previously reported neutralizing and non-neutralizing S2-directed antibodies that target the highly conserved stem helix (IgG22[2], S2P6[23] and CC40.8[25]), fusion peptide (CoV44-79 and CoV91-27[24]), and apex (B3-1[64,65]) against HexaPro-SS (Fig. 6c), HexaPro-SS-2W (Fig. 6c), and HexaPro-SS-T998W (Supplementary Fig. 25) constructs (including the stalk region). An antibody targeting the RBD of S1 was used as a control (N3-1[65]). BLI shows that the binding affinity of antibodies that target the stem helix or fusion peptide is not affected in HexaPro-SS-2W when compared to HexaPro-SS. However, the binding affinity of the non-neutralizing antibody B3-1 targeting the apex of the S2 trimer as in HexaPro-SS-2W was reduced by half with respect to HexaPro-SS (Fig. 6c). This suggests that the opening of the S2 apex is important for B3-1 binding. Interestingly, B3-1 binding is not fully abrogated, suggesting that HexaPro-SS-2W may infrequently undergo opening motions, as shown from WE simulations.

## Discussion

In this work, we present the simulation-driven design of SARS-CoV-2 S2-only antigens stabilized in the closed prefusion conformation. We first used WE path-sampling MD simulations to investigate the conformational plasticity of HexaPro-SS-Δstalk, a prefusion-stabilized S2 construct that exhibits flexibility at the apex, leading to splayed open conformations[32]. We discovered that the prefusion-stabilized S2 trimer transitions from a closed to open conformation through an asymmetric protomer-protomer separation. Contact map analysis of the interfacial helices (CHs, UHs, and HR1) reveals key residue–residue interactions taking place in the opening pathway. To improve interprotomer stabilization, we identified two residues located in the CH apex, V991 and T998, as candidates for cavity-filling substitutions with tryptophan. We next designed three variants of HexaPro-SS-Δstalk incorporating V991W, T998W, and V991W + T998W substitutions, namely HexaPro-SS-V991, HexaPro-SS-T998, and HexaPro-SS-2W. Additional WE simulations of the three newly engineered S2 variants showed that T998W and V991W + T998W provide kinetic stabilization by slowing down the S2 trimer opening. Interestingly, when only position 991 is substituted to tryptophan (i.e., in HexaPro-SS-V991W), it does not induce slower opening kinetics despite furthering

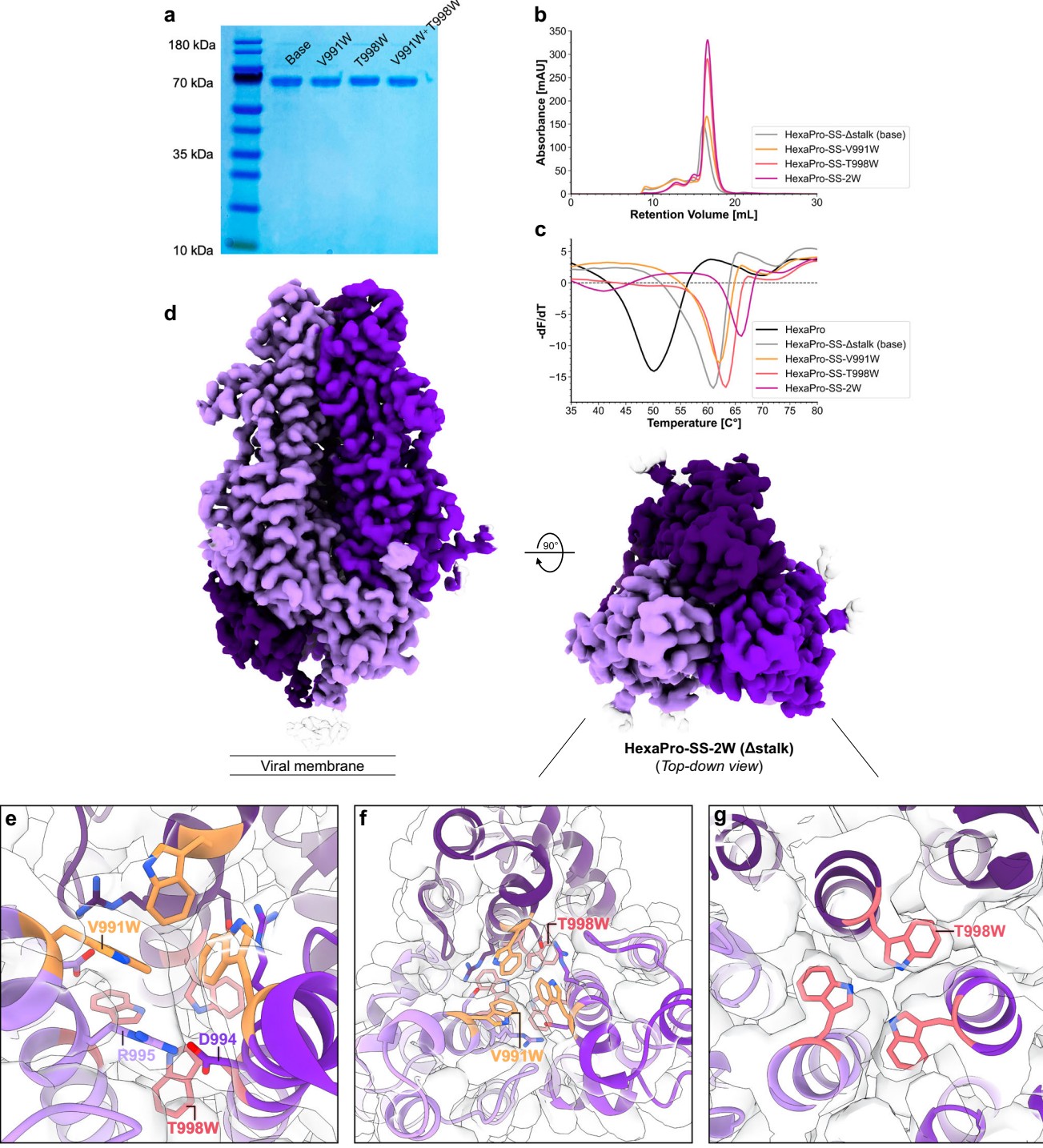

**Fig. 5 | Cellular expression, thermostability, and structural characterization of HexaPro-SS-2W. a** SDS-PAGE of purified S2 constructs (Base, HexaPro-SS-V991W, HexaPro-SS-T998W, and HexaPro-SS-2W). The 'Base' construct corresponds to HexaPro-SS-Δstalk. No statistical analysis was performed on the SDS-PAGE gel. The uncropped gel image is included in the Source Data file. **b** Size-exclusion chromatography (SEC) of purified S2 constructs. Both tryptophan substitutions increase protein expression yield, with V991W + T998W and T998W resulting in a greater increase. **c** Differential scanning fluorimetry of S2 constructs, including the original HexaPro (S1 + S2). HexaPro-SS-2W exhibits superior thermal stability to all HexaPro constructs, with a -16 °C increase in $T_m$ relative to HexaPro. **d** Cryo-EM map (Resolution: 2.8 Å) of closed prefusion state HexaPro-SS-2W (Δstalk) from side and top-down perspectives. **e–g** Top-down perspective of HexaPro-SS-2W structure highlighting V991W (orange) and T998W (red) packing within the S2 interior; the tryptophan sidechains self-associate between chains in offset edge-to-edge and edge-to-face π–π stacking orientations. The EM map is shown as a transparent gray volume.

interprotomer engagement. This could be ascribed to the apical position of W991, which is more solvent-exposed than W998 and, in the absence of a partner hydrophobic residue underneath (i.e., W998 in HexaPro-SS-2W), is not kinetically stabilizing during the opening transition.

Subsequent non-equilibrium alchemical free energy calculations revealed a negative $\Delta\Delta G_{mutation\text{-}folding}$ imparted by the tryptophan substitutions, evincing increased stability of the closed prefusion conformation for all three S2-engineered variants, with HexaPro-SS-2W exhibiting superior stability. Strikingly, cellular expression and

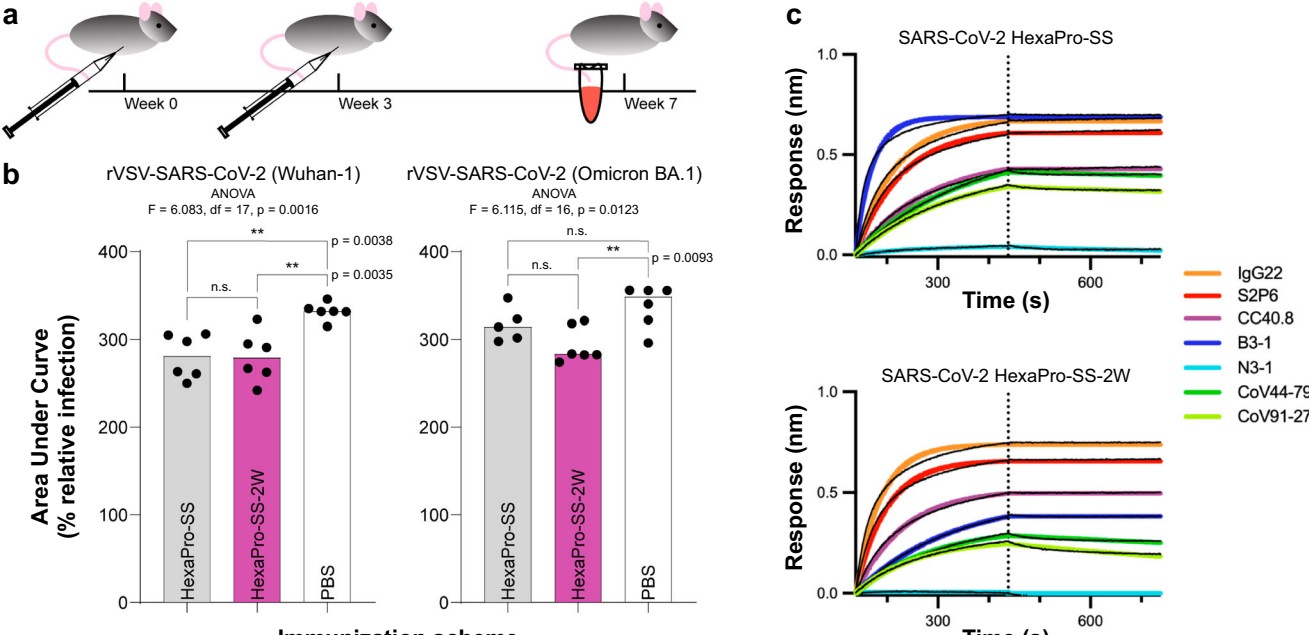

**Fig. 6 | Immunogenicity of HexaPro-SS and HexaPro-SS-2W constructs. a, b** Sera from mice immunized with HexaPro-SS-2W can neutralize rVSV-CoV-2 variants. **a** 6–8-week-old female C57BL/6J mice (*n* = 6) were primed (week 0) and boosted (week 3) with 10 μg of immunogens HexaPro-SS, HexaPro-SS-2W, or PBS. **b** Sera was isolated at week 7 and evaluated for neutralization capacity against rVSVs bearing the spike proteins of SARS-CoV-2 Wuhan-1 or SARS-CoV-2 Omicron BA.1. Lower area under the curve (AUC) values correspond to better sera neutralizing capability. AUCs were compared across groups by ordinary one-way ANOVA with Tukey's multiple comparisons test (** indicates a *p*-value < 0.01; n.s. indicates not significant). **c** Biolayer interferometry sensorgrams showing binding responses (colored lines indicate 1:1 binding fit, black lines indicate reference subtracted response) of previously reported neutralizing or non-neutralizing antibodies to epitopes on the S2 subunit in HexaPro-SS (top) and HexaPro-SS-2W (bottom) constructs.

thermostability characterization of the newly stabilized S2 constructs confirmed the computational predictions, showing higher expression yields and melting temperatures for the three tryptophan variants, with HexaPro-SS-2W surpassing all the other constructs. The improved stability of HexaPro-SS-2W allowed the high-resolution characterization of its closed prefusion structure via cryo-EM. The resolved cryo-EM structure highlights key residue interactions also observed in the WE simulations. As these interactions in the closed state are not protomer-specific, this suggests that the synergistic residue coordination in the S2 interior can dynamically exchange interactions to stabilize a closed conformation. Importantly, immunogenicity assays on sera from mice immunized with HexaPro-SS-2W demonstrated that this stabilized S2-only antigen can neutralize different rVSV-SARS-CoV-2 variants. Furthermore, BLI assays showed that the binding of the non-neutralizing antibody B3-1 to the HexaPro-SS-2W's apex is reduced by half when compared to the base construct. Reducing the binding affinity of the non-neutralizing antibodies targeting S2 may indicate that closing the S2 trimer with 2 W can bias immunization responses towards neutralizing epitopes such as stem helix and fusion peptide as opposed to non-neutralizing epitopes such as the apex. Overall, further testing is needed to assess the neutralizing response across a broader panel of beta-coronaviruses, including MERS-CoV, HKU-1, OC43, SARS-CoV, and additional SARS-CoV-2 variants. Protection data will further illuminate the potential of the HexaPro-SS-2W antigen as a vaccine candidate.

Our approach underscores the advantage of using dynamic information from molecular simulations to inform immunogen design. Static-structure methods depend on energetic calculations of protein structures in cryogenic conditions or rely on the structural homology of proteins with minor conformational variance. In contrast, our simulations capture dynamic information from an ensemble of conformations at reproduced physiological conditions. The analysis of the representative conformational ensemble can help decipher the residue-specific contributions to conformational stability, shedding light on the impact of mutations in emerging SARS-CoV-2 VOCs. Given the rapid accumulation of sequence modifications in the spike protein, investigating the effects of these mutations may become crucial in designing robust vaccines against future variants. Our approach, or alternative MD-based approaches such as mesoscale simulations[33], algorithms to study allostery[66–68] and epitope prediction[69–71] could also be used to characterize short-lived antigenic conformations, dissect the antigenic impact of escape mutations, or reveal cryptic epitopes. Considering the challenges to obtain atomic-level details of conformational ensembles from experiments, the mechanistic insights gained from computational methods could play a pivotal role in steering vaccine design efforts. The computational approach used here enabled the design of stabilized SARS-CoV-2 S2-only immunogens that could serve as the structural basis for future pan-coronavirus vaccine development efforts and may be generally applicable to structure-based vaccine design.

## Methods

### Ethical statement

This research complies with all relevant ethical regulations; the study protocol was approved by the Albert Einstein College of Medicine Institutional Animal Care and Use Committee (protocol no: 00001535).

### Modeling of the closed prefusion HexaPro-SS-Δstalk S2 trimer

The all-atom model of the closed prefusion HexaPro-SS-Δstalk S2 trimer, spanning residues 696–1141, was constructed from a cryo-EM structure of the SARS-CoV-2 HexaPro spike (PDB ID: 6XKL[31]). The fusion peptide region (residues 823–857) was grafted from a cryo-EM structure of the SARS-CoV-2 3Q-2P spike (PDB ID: 7JJI[41]). The HexaPro-SS-Δstalk stabilizing mutations (S704C, K790C, and Q957E) were incorporated with VMD's psfgen 2.0 structure-building tool[42]. The

model was glycosylated using the same glycoprofile from Casalino et al[11]. and parameterized using the all-atom additive CHARMM36m force field for protein and glycans[72–74]. The system was solvated with TIP3P[75] explicit water molecules in a box with at least 15 Å between the protein and box edges (box size: 134 × 140 × 153 Å), and the charge was neutralized with Na$^+$ and Cl$^-$ ions up to a concentration of 150 mM. The final system accounted for 274,082 atoms.

## All-atom MD simulations of HexaPro-SS-Δstalk
Conventional all-atom MD simulations of the closed prefusion HexaPro-SS-Δstalk system were performed using NAMD3[76]. An energy minimization was initially performed for 10,800 steps utilizing the conjugate gradient method. During this phase, protein and glycan atoms were subjected to positional harmonic restraints, applying a force constant of 1.0 (kcal/mol)/Å$^2$. Following this, a 0.5 ns NPT (isothermal-isobaric) equilibration was conducted. The harmonic restraints were retained throughout this stage. Next, restraints were released, and an additional 5 ns of NPT equilibration was performed. Subsequently, four replicates of NPT production MD simulations were run (R1 = 601.5 ns, R2 = 655.4 ns, R3 = 744.6 ns, R4 = 627.6 ns) after re-initializing velocities. Control of the temperature and pressure was achieved with a Langevin thermostat[77,78] (310 K) and a Nosé-Hoover Langevin barostat[79] (1.01325 bar). All simulations (including equilibration) were performed using an integration time step of 2 fs and the SHAKE[80] algorithm to keep covalent bonds involving all hydrogen atoms fixed. Periodic boundary conditions were active, with the particle-mesh Ewald[81] method used for long-range electrostatics calculation. Non-bonded interactions, such as van der Waals interactions and short-range electrostatics, were accounted for with a cut-off of 12 Å.

## WE MD simulation of HexaPro-SS-Δstalk
The initial HexaPro-SS-Δstalk construct for WE simulations was extracted from a frame parsed from conventional MD simulations. The dry construct was re-solvated with TIP3P[75] water molecules in an orthorhombic box of 138 × 150 × 168 Å per side, allowing at least 24 Å between the solute and the box edges in all directions, and neutralized with a 150 mM concentration of NaCl. As a result, we obtained a system tallying 328,200 atoms. In order to use the Amber software[50] as the engine for the subsequent WE MD simulations while maintaining the CHARMM36m force fields, we used the CHAMBER[82] program to convert the generated coordinate (.pdb) and topology (.psf) files, and the necessary CHARMM36m parameters files into an Amber readable format (i.e., .prmtop file for topology and parameters and .rst7 file for coordinates). To ensure a more extensive sampling of the initial state in a WE simulation, it is recommended to provide a set of initial, equally weighted conformations (called 'basis states') rather than a single conformation. To accomplish this task, we performed 10 preparatory MD simulations of 50 ns each using Amber20 software[50] and CHARMM36m force fields[72–74]. The full protocol is described in the Supplementary information. From the generated ensemble of closed conformations, we extracted 50 initial basis states, to which we assigned an initial weight of 0.02 each. Criteria used for the selection of basis states are fully detailed in the Supplementary information.

All WE MD simulations were performed using WESTPA 2.0 software[48], exploiting Amber20 PMEMD.cuda[50] as MD engine to propagate dynamics. CHARMM36m force fields[72–74] were adopted for protein and glycan, and TIP3P[75] for water molecules. To enhance sampling of closed-to-open transitions of HexaPro-SS-Δstalk S2 trimer, we defined two progress coordinates. The first progress coordinate was set as the area of the triangle having the Cα atom of residues P987$_A$, P987$_B$ and P987$_C$ as vertices. The second progress coordinate was set as the RMSD of the Cα atoms of the CHs (residues 987–1033) in the simulation construct to the corresponding Cα atoms of the CHs (residues 987–1033) in the HexaPro-SS-Δstalk open crystal structure

(PDB ID: 8U1G[32]). The minimal adaptive binning (MAB) scheme was applied to bin the conformational space defined by the two progress coordinates[83]. The resampling time interval τ was set to 100 ps, whereas the target number of independent trajectories run in each bin, M, was set to 8. MD simulations were performed in NPT conditions using a Langevin thermostat[77,78] to maintain the temperature at 300 K and the Monte Carlo barostat[84] to keep the pressure at 1 bar. SHAKE algorithm was used to constrain bonds involving hydrogen atoms[80]. Particle Mesh Ewald method was used to handle long-range electrostatic interactions[81], whereas non-bonded interactions were calculated with a cutoff of 10 Å. WE simulation of HexaPro-SS-Δstalk was carried out for a total of 315 iterations, ~46 μs aggregate sampling, and 180 successful opening pathways (Supplementary Fig. 12 and Supplementary Tables 1, 3). In order to be able to provide an estimate of kinetic rates for the opening transition, we facilitated the convergence of WE simulations toward near equilibrium steady-state conditions by applying the WE Equilibrium Dynamics (WEED) protocol twice to reweight the trajectories[56,58]. Convergence of WE simulations was assessed by monitoring the evolution of the 1D probability distributions relative to each progress coordinate (Supplementary Figs. 2b, 8b, 9b, 10b) as well as by tracking the evolution of the kinetic rates' rolling averages calculated for the opening transition (Supplementary Fig. 15). A complete description of system setup, progress coordinate selection, state definition, and WE simulation protocol is provided in the Supplementary information.

## Modeling and WE MD simulations of HexaPro-SS-V991W, HexaPro-SS-T998W, and HexaPro-SS-2W variants
The three HexaPro-SS-Δstalk tryptophan-stabilized systems (HexaPro-SS-V991W, HexaPro-SS-T998W, and HexaPro-SS-2W) were built from the same base construct used in the HexaPro-SS-Δstalk WE simulation. V991W, T998W, or V991W + T998W substitutions were incorporated in each protomer of the respective HexaPro-SS-Δstalk variant using the psfgen 2.0 structure-building tool within VMD[42]. A similar system setup and WE simulation settings, as described for HexaPro-SS-Δstalk, were used for each variant. WE simulation of the HexaPro-SS-V991W mutant was carried out for a total of 301 iterations, ~36 μs aggregate sampling, and 149 successful opening pathways (Supplementary Fig. 12 and Supplementary Tables 1, 3). WE simulation of the HexaPro-SS-T998W mutant was performed for a total of 355 iterations, ~48 μs aggregate sampling, and 187 successful opening pathways (Supplementary Fig. 12 and Supplementary Tables 1, 3). WE simulation of the double HexaPro-SS-2W mutant was carried out for a total of 401 iterations, ~57 μs aggregate sampling, and 195 successful opening pathways (Supplementary Fig. 12 and Supplementary Tables 1, 3). For all the mutants, trajectory reweighting was conducted twice using the WEED plugin of WESTPA 2.0[56,58]. A complete description of the system setup and WE simulation protocol is provided in the Supplementary information.

## Simulation analysis
WE simulations were analyzed using analysis tools provided by WESTPA 2.0[48] software, including the estimation of kinetic rates. Contact map analysis of the opening pathways was performed with PyContact[51]. Contact analysis for the closed conformations was performed with MDAnalysis[85]. Images and movies relative to the simulations were generated with VMD[42]. A complete description of all the analyses, including analysis settings and step-by-step protocols, is provided in the Supplementary information.

## Non-equilibrium alchemical free energy calculations
To build the thermodynamic cycle (Supplementary Fig. 16) for non-equilibrium alchemical free energy calculations, two independent systems representing the unfolded and folded states of the protein were generated. To approximate the unfolded state, considering the limitations of empirical force fields in simulating unfolded proteins[86–88]

and given the unavailability of the S2 trimer unfolded structure, we used a capped tripeptide (G-X-G) built with Chimera[89], where X is the amino acid of interest (either V, T, or W). We note that this approach has been commonly adopted in alchemical free energy calculations and has proven to align well with experimental data[62,90,91]. For the folded state, wild-type system, we used the same equilibrated closed HexaPro-SS-Δstalk construct in the closed conformation extracted from conventional MD. Mutant systems for the folded state (HexaPro-SS-V991W, HexaPro-SS-T998W, HexaPro-SS-2W in the closed conformation) were constructed from the HexaPro-SS-Δstalk base construct using the psfgen 2.0 structure-building tool within VMD[42]. For each variant, mutations were initially incorporated into one protomer only (protomer A), generating the respective A-mutated construct. Next, starting from this construct, a second system was built by introducing the respective mutation also in the second protomer (protomer B). Finally, a third construct incorporating the respective mutation in all three protomers (A, B, and C) was created. Free energy calculations were then performed sequentially, and the total change in free energy for the mutated trimer was recovered by combining the $\Delta G$ values for each independent calculation. The PMX[90] software was used to build hybrid structures and topologies incorporating sidechains from both the wild-type and mutant amino acids. Generated structures of the solute were then solvated in a cubic TIP3P[75] water box using Gromacs 2022.1 software[92], imposing at least 12 Å from the solute edges in each direction. Na$^+$ and Cl$^+$ ions were added at a concentration of 150 mM to neutralize the systems. For both wild-type and mutant systems in both the folded and unfolded states, equilibrium MD simulations were carried out independently for 20 ns in the NPT ensemble at a target pressure of 1 bar using the Parrinello-Rahman barostat[93] and a time step of 2 fs. Equilibrium simulations were performed using Gromacs 2022.1 software[92] and CHARMM36m force fields[72,73] and TIP3P[75] model for water molecules. The full simulation protocol is described in the Supplementary information. In order to aid the convergence of the subsequent non-equilibrium MD simulations, the aforementioned procedure was repeated five times for both states (folded and unfolded) of each mutant system. Frames were saved every 100 ps, corresponding to 200 frames saved per simulation. These frames were used to spawn a total of 1000 non-equilibrium MD simulations in the forward direction (wild-type-to-mutant) and 1000 non-equilibrium MD simulations in the reverse direction (mutant-to-wild-type) for both the folded and the unfolded state. The alchemical transition between wild-type and mutant was achieved in 100,000 steps (200 ps) by switching the λ parameter from 0 to 1 or from 1 to 0, depending on the direction of the transition, using a gradient of 0.00001/step[62]. Convergence was assessed by evaluating the intersection between the work values calculated from the forward and reverse non-equilibrium simulations (Supplementary Fig. 17). During these simulations, the soft-core function and the default parameters for Coulombic and van der Waals interactions were used[94]. PMX[90] *analyse* script was used to extract the work values corresponding to each alchemical transition, and the free energy differences were calculated based on the Crooks fluctuation theorem[95] by utilizing the Bennett acceptance ratio (BAR) as a maximum likelihood estimator[96]. A thermodynamic cycle was then constructed, as shown in Supplementary Fig. 16, and the difference in protein folding free energy upon mutation ($\Delta\Delta G_{mutation\text{-}folding}$) was calculated. A full description of non-equilibrium alchemical free energy calculations is provided in the Supplementary information.

## Protein expression

V991W, T998W, and V991W/T998W substitutions were introduced into HexaPro-SS (SARS-CoV-2 HexaPro S, encompassing amino acid residues 697–1208, with S704C, K790C, and Q957E substitutions) and HexaPro-SS-Δstalk (encompassing amino acid residues 697–1141, with S704C, K790C, and Q957E substitutions) constructs containing a

C-terminal foldon trimerization motif, an HRV 3C cleavage site, 8x His tag, and Twin-Strep-tag[32]. Plasmids (0.125 mg) encoding HexaPro-SS, HexaPro-SS-Δstalk, and the tryptophan variants were transiently transfected into 250 mL FreeStyle™ 293F cells (Thermo Fisher, cat #R79007) with 25 kDa linear polyethylenimine (PEI) (Polysciences cat# 3966-2), and sterile-filtered kifunensine was added 3 h post-transfection to a final concentration of 5 μM. The medium was harvested and filtered 6 days post-transfection. StrepTactin Sepharose resin (IBA, cat #2-1201-002) was used for affinity purification, and the elution fraction was applied onto a Superose™ 6 Increase 10/300 GL (Cytiva, cat #29091596) size-exclusion chromatography (SEC) column in a running buffer of 2 mM Tris pH 8.0, 200 mM NaCl, and 0.03% NaN₃. Peak fractions were eluted, combined, and concentrated using 30K Amicon Ultra centrifugal filters. Endotoxin-free protein samples for mouse immunizations were prepared using Pierce™ High Capacity Endotoxin Removal Spin Columns (Thermo Scientific, cat # 88274).

Antibody IgGs N3-1 and B3-1 were gifted from the Jimmy Ghol-lihar lab at Houston Methodist Research Institute. Antibody IgGs IgG22, S2P6, CC40.8, CoV44-79, and CoV91-27 were cloned from codon-optimized VH and VL gBlock gene fragments (Integrated DNA Technologies IDT) into human IgG1 heavy chain and light chain backbones. Antibody plasmids (0.25 mg heavy chain and 0.25 mg light chain) were co-transfected into 1000 mL FreeStyle™ 293F cells (Thermo Fisher, cat #R79007) using the same method as described above. The medium was harvested and filtered 6 days post-transfection. Protein A Plus Agarose (Thermo Scientific, PI22812) was used for affinity purification, and the elution fraction was applied onto a Superose™ 6 Increase 10/300 GL (Cytiva, cat #29091596) SEC column in a running buffer of 2 mM Tris pH 8.0, 200 mM NaCl, and 0.03% NaN₃. Peak fractions were eluted, combined, and concentrated using 30K Amicon Ultra centrifugal filters.

## Cryo-EM sample preparation

The SARS-CoV-2 HexaPro-SS-2W (Δstalk) construct was diluted to 0.8 mg/mL in 2 mM Tris pH 8.0, 200 mM NaCl, 0.02% NaN₃. A 4 μL aliquot of the sample was placed onto a glow discharged UltrAuFoil R 1.2/1.3 300 mesh gold grid (Electron Microscopy Sciences, Cat. # Q350AR13A). The grid was then subjected to a 5-s wait time in a 22 °C, 100% humidity chamber, blotted for 4 s with −3 force, and plunge-frozen into liquid ethane using a Vitrobot Mark IV (Thermo Fisher). A total of 1764 micrographs were collected on a 200 kV Glacios (Thermo Fisher) equipped with a Falcon 4 detector. Data were collected at 150,000× magnification, corresponding to 0.94 Å/pix. A full description of the cryo-EM data collection can be found in Supplementary Table 4.

## Cryo-EM data processing and model building

CryoSPARC[97] Live v4.0.2 was used for motion correction, CTF estimation, micrograph curation, and particle picking. Particles were picked using a template picker with an extraction box size of 384 (pix). The selected particles and accepted micrographs were exported into CryoSPARC v4.0.2 for ab initio 3D reconstruction, heterogeneous refinement, and homogeneous refinement. C3 symmetry was applied in the final reconstruction. The C3 map was further sharpened using DeepEMhancer[63] via the Cosmic2[98] science gateway. ChimeraX[89], Isolde[99], Phenix (real-space refinement)[100,101] and Coot[102] were used interactively to build the model. A full description of cryo-EM data processing can be found in Supplementary Fig. 22.

## Differential scanning fluorimetry

Solutions containing a final protein concentration of 1 μM and 6.25X SYPRO Orange Protein Gel Stain (Thermo Fisher, cat #S5692) were added to the wells of a 96-well qPCR plate. Continuous fluorescence measurements (λex = 465 nm, λem = 580 nm) were performed using a Roche LightCycler 480 II at a temperature ramp rate of 4.4 °C/min

increasing from 22 to 95 °C. Data were plotted as the negative of the derivative of fluorescence as a function of temperature.

### Biolayer interferometry

Anti-hIgG Fc Capture (AHC) Biosensors (Sartorius, cat # 18-5064) were pre-wet in 1X HBS-EP+ buffer (Cytiva, Cat # BR100669) for 10 min. IgGs (with IgG G4[30] as an isotype control) were diluted in the same buffer to 25 nM at a total volume of 200 μL per well. HexaPro-SS and tryptophan-stabilized constructs were diluted in the 1X HBS-EP+ buffer to a final concentration of 100 nM in 200 μL per well. The AHC biosensors first underwent 60 s of baseline in buffer and were then dipped into IgG wells for capture to a level of 0.6 nm on Octect RED96e (FortéBio). The captured biosensors were then transferred back to buffer wells for another 60 s of baseline and then dipped into wells containing spike protein constructs for a 300-s association, followed by transfer into wells containing buffer for another 300-s dissociation. Data were then reference subtracted (isotype control) and fit for both association and dissociation to a 1:1 binding model using Data Analysis 11.1 software.

### Cell culture

Vero cells (ATCC) were cultured in Dulbecco's modified Eagle's medium (DMEM, high glucose; Gibco) supplemented with 2% heat-inactivated fetal bovine serum (FBS; Bio-Techne), 1% penicillin-streptomycin (P/S; Thermo Fisher Scientific), and 1% GlutaMAX (Thermo Fisher Scientific). 293FT cells (Thermo Fisher Scientific) were cultured in DMEM supplemented with 10% FBS, 1% P/S, and 1% GlutaMAX.

### Generation of rVSV-SARS-CoV-2 viruses

Replication-competent vesicular stomatitis virus (rVSV) pseudotype viruses bearing the CoV spike of interest was previously described by Dieterle et al.[103]. Briefly, a plasmid encoding the VSV antigenome was engineered to replace its native glycoprotein G with the CoV spike of interest. The spike sequence of rVSV-SARS-CoV-2 Wuhan-1 was obtained from GenBank (MN908947.3), while that of rVSV-SARS-CoV-2 Omicron was generated by incorporating the defining mutations of B.1.1.529 (as reported by CoVariants[104]) in the Wuhan sequence. DNA fragments for both sequences were produced by Twist Bioscience. As previously reported, a 19 amino acid C-terminal deletion was made in each CoV spike. An enhanced green fluorescent protein (eGFP) reporter gene was included in the VSV antigenome as a separate transcriptional unit. Plasmid-based rescue of the rVSVs was carried out as previously described[103,105]. Briefly, 293FT cells were transfected using polyethyleneimine with the VSV antigenome plasmid along with helper plasmids expressing T7 polymerase and VSV proteins: N, P, M, G, and L. Supernatants from transfected cells were transferred to Vero cells 48 h post-transfection. The appearance of eGFP-positive cells confirmed the presence of virus. Viral clones were plaque-purified on Vero cells and propagated by cell subculture. RNA was isolated from viral supernatants, and Sanger sequencing was performed to confirm the spike gene sequence. Viral stocks were concentrated via ultracentrifugation in an SW28 rotor at 141,000×g for 4 h. Virus was aliquoted and stored at −80° C. The generation of all rVSV-CoVs and their use in tissue culture was done at biosafety level 2 and was approved by the Environmental Health and Safety Department and Institutional Biosafety Committee at Albert Einstein College of Medicine.

### In vivo immunization

Six- to 8-week-old female C57BL/6J mice were purchased from The Jackson Laboratory (JAX: 000664). Mice ($n = 6$) were immunized with 10 μg of each S2 antigen construct or PBS adjuvanted with 10 μg of Sigma Adjuvant System (Sigma) and boosted 3 weeks after in the same fashion. Constructs and adjuvant were each diluted in endotoxin-free PBS (Millipore) to a total volume of 100 μL before they were mixed to obtain a total of 200 μL per mouse. Then, 200 μL of immunogen/adjuvant were delivered by intraperitoneal injection for each mouse. Mice were bled 1 week prior to prime (week −1), boost (week 2), and 4 weeks post-boost (week 7). Serum was collected from each mouse, heat-inactivated for 30 min at 56 °C, aliquoted, and stored at −80 °C for subsequent use in rVSV-CoV neutralization assays.

### rVSV-CoV neutralization assays

Serial 3-fold dilutions were made for each mouse serum sample starting at a 1:10 dilution in DMEM supplemented with 2% FBS, 1% P/S, and 1% GlutaMAX. Mouse sera was incubated with rVSV-SARS-CoV-2 (Wuhan or Omicron) for 1 h at room temperature. Media was removed from Vero cells in a 96-well cell culture plate (Corning 3585) and 40 μL virus/sera mixture was added to well in singlicate. The cells were incubated with the virus/sera mixture at 37 °C and 5% $CO_2$ for 10 h. The cells were then fixed with 16% paraformaldehyde (PFA; Thermo Fisher Scientific) to a final PFA concentration of 4%, washed with 1x PBS and stained for cell nuclei in 1x PBS with Hoechst 33342 (Invitrogen) at a dilution of 1:2000. Viral infection was quantified by automatic enumeration of GFP-positive cells from captured images using Cytation 5 automated fluorescence microscope (BioTek) with analysis done by Gen5 (version 3.12) data analysis software (BioTek). The area under the curve (AUC) of sera sample was calculated using GraphPad Prism software (Version 9.3.1) with the following parameters: baseline Y = 0; minimum peak height less than 0 Y units high. Each sera group was evaluated for normality and homoscedasticity by QQplot and Bartlett's test, respectively. Ordinary one-way ANOVA with Tukey's multiple comparisons test was performed for each neutralization study. All statistical testing was performed using GraphPad Prism software (Version 9.3.1).

### Reporting summary

Further information on research design is available in the Nature Portfolio Reporting Summary linked to this article.

## Data availability

Trajectories of S2 trimer opening pathways, WE configuration files, and representative closed prefusion conformations of HexaPro-SS-2W are included in the article as Supplementary Data 1. Full datasets (simulation input files and trajectories) for conventional MD, GaMD, WE MD, and free energy calculations are available for download on the Amaro Lab website [https://amarolab.ucsd.edu/covid19.php] under the name 'HexaPro-SS_S2_cMD_GaMD_WE_FEC_simulation_data.tar.gz.' The atomic coordinates of the open prefusion HexaPro-SS-Δstalk crystal structure referenced in this work are available in the Protein Data Bank (PDB) under accession code 8U1G. The cryo-EM maps of HexaPro-SS-Δstalk-2W in the closed prefusion conformation generated in this study have been deposited in the Electron Microscopy Data Bank (EMDB) under accession codes EMD-43097. The atomic coordinates of HexaPro-SS-Δstalk-2W in the closed prefusion conformation generated in this study have been deposited in the Protein Data Bank (PDB) under accession codes 8VAO. Source data are provided with this paper.

## Code availability

This study utilized the standard builds of the simulation software WESTPA 2.0 (https://github.com/westpa/westpa), AMBER 20 (https://ambermd.org), NAMD 2.14/NAMD 3 (https://www.ks.uiuc.edu/Research/namd/), PMX, and Gromacs 2022.1 (https://manual.gromacs.org/2022.1/release-notes/2022/2022.1.html) according to best practices with no special modifications.

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

## Acknowledgements

We thank Dr. Lillian T. Chong and Dr. Daniel Zuckerman for helpful discussions about WE simulations. L.C. thanks Dr. Anthony Bogetti for the support with WESTPA software. M.S. thanks Dr. Vytautas Gapsys for the support with free energy calculations. All the simulations were run on the Triton Shared Computing Cluster (TSCC) at the San Diego Supercomputer Center (SDCC). We thank them for their support and assistance. This work was funded in part by the Bill & Melinda Gates Foundation INV-031624 (K.C., J.S.M., R.E.A.), by the Welch Foundation grant number F-0003-19620604 (J.S.M.), by the National Institute of Health Medical Scientist Training Program T32GM007288 (A.W.), and by the UC San Diego Foundation Chancellor Funded Research Grant (R.E.A.). X.N. acknowledges support from the Molecular Biophysics Training Grant T32 GM139795. S.A. acknowledges support from the Department of Chemical Engineering at the University of California, Davis.

## Author contributions

X.N. and L.C. designed the S2 cavity-filling substitutions (V991W and T998W). X.N. and L.C. prepared all the simulation models. X.N. designed and performed conventional MD and GaMD simulations. L.C. designed and performed WE simulations. X.N. and L.C. designed and performed all simulation analyses. M.S., A.A.O., F.L.K., M.A.R., M.S., and R.E.A. critically helped design simulation analyses. M.S. designed and performed free energy calculations and related analyses and figures. X.N. and L.C. made all the figures and movies. S.A. provided guidance with the WESTPA setup and simulation scripts. A.A.O. helped configure and design WE progress coordinates. R.E.A. designed and oversaw all the simulations. L.Z. designed and performed DNA cloning, expression tests, expression for mouse studies, cryo-EM, and BLI experiments, and made related figures. C.M.A. performed expressions and DSF experiments. J.S.M. designed and oversaw all the experiments. A.W., A.L.T., and E.H.M. designed and performed mouse immunizations, rVSV-SARS-CoV-2 viral rescues, and neutralization assays, and made related figures. K.C. designed and oversaw immunogenic and viral assays. R.E.A. oversaw the project. X.N. and L.C. wrote the original draft of the manuscript, and all the authors contributed to reviewing and editing.

## Competing interests

J.S.M., L.Z., R.E.A., X.N., L.C., and M.S. are inventors on a U.S. patent application describing the use of stabilized SARS-CoV-2 S proteins as vaccine antigens (Stabilized SARS-CoV-2 S Antigens, 63/583,090). K.C. is a member of the scientific advisory board of Integrum Scientific LLC and has consulted for Axon Advisors, LLC. K.C. owns shares in Integrum Scientific and Eitr Biologics, Inc. The other authors declare no competing interests.
