## [Peer Review File · Nature Communications]

Simulation-Driven Design of Stabilized SARS-CoV-2 Spike S2 ImmunogensREVIEWER COMMENTS

Reviewer #1 (Remarks to the Author):

This is a very interesting, up to date, and exciting paper on the redesign of stabilized S2 variants of the Sars-CoV-2 spike protein.

The Authors combine long classical MD simulations with WE analysis to profile the opening mechanism of the protein.

The data are used to infer mutations able to fill voids in S2, with the working hypothesis of stabilizing this domain.

The authors validate their design with Cryo-EM, thermostability and expression data. Importantly, stabilized variants show enhanced immunogenicity also against distinct variants.

I found this paper very nice and interesting to read and I find it suitable for Nat Comm. The authors might consider expanding even more the generality of their discussion and implications by discussing their data in the context of integrating methods to discover new epitopes against distinct variants, or to integrate other types MD-based analyses to characterize stability.

Reviewer #2 (Remarks to the Author):

In this article, Nuqui et al. present the design of a variant of the subunit S2 of the SARS-CoV-2 spike protein with great potential as immunogen for vaccine development that could address the inefficacy towards the emerging virus variants of vaccines based on the less conserved subunit S1. The design is carefully described and originated by a thorough computational investigation using a number of advanced calculations. The in silico results are supported by experiments, including the resolution of a Cryo-EM structure and integrated by both in vivo and in vivo assays. The work is technically solid, clearly written, and the conclusions supported by the results. These aspects together with the relevance of the discovery make the manuscript of great interest and I suggest its publication in Nature Communications.

A number of points to improve the quality of the work are reported below.

- A cartoon representation of the functional motion of the Spike protein, reported as Fig. 1, would be beneficial for the reader;

- Page 6, line 24 - the fusion peptide's residues should be highlighted in the original Fig. 1;

- Figure 3 - It would be interesting to show the contact values as function of the simulation time. This analysis might provide insight into the opening mechanism of S2.

In addition, the partially open and the open conformation of S2 should be shown in this figure together with the closed one in order to appreciate the structural differences;

- Why the transition from the closed to the open states of S2 is barrierless in the probability plots shown in Fig. S2A, S7A, S8A and S9A. Especially in the HexaPro-SS-2W, considering the thermodynamic stability of the closed state in this system, I would expect a relatively large barrier separating the two states. That would also explain the slower rate computed for the opening motion in such a system with respect to the other systems. Could the authors explain this unexpected result?

- Thermodynamic and kinetic estimates obtained from WE simulations rely on the convergence of the calculations. If an investigator badly chooses the convergence point of the simulations, the obtained results might be incorrect. Considering the relevance of this point, the authors should include at least in the Methods section, and in the SI, how they evaluate the convergence of the WE simulations.

Moreover, the authors wrote in different parts of the manuscript and SI "To facilitate the convergence of WE simulations toward equilibrium, we applied the WE Equilibrium Dynamics (WEED) protocol twice to reweight the trajectories". What does this exactly mean? Reweighting is used to recompute the unbiased Boltzmann distribution (probability function), but if the calculations is not converged, even the reweighted probability is not converged. A possible explanation is that the probability plots reported in Fig. S2B and (less in S8B and S9B) show similar profiles computed at different simulation times, suggesting an approach to the convergence of the calculations. The authors should however clarify this point;

- Page 24, line 13 - "For the unfolded state, we used a capped tripeptide (G-X-G) built with Chimera, where X is the amino acid of interest (either V, T, or W). I do not understand why the authors simulated the tripeptide G-X-G as unfolded state to compute the free-energy difference between the folded and unfolded state (alchemical free energy calculations). That tripeptide is not representative of the real unfolded state of S2 and the estimates might be biased by this choice. A natural choice would have been to use the open state as reference to computed the free-energy difference with respect to the closed state (not folded state). This calculation would further provide an orthogonal validation of the WE calculations. The authors should motivate in the main text the reason why they opted for the tripeptide G-X-G;

- The resolution of the Cryo-EM structure should be reported. Did the authors try to resolve at Cryo-EM the structure of the HexaPro-SS- Δ stalk S2 (wild type)? It would be interesting a comparison;

- Page 17, line 24 - This part is not clear in my opinion. The decreased infectivity seems to be not so strong looking at the histograms in Figure 6B, also considering the uncertainty (errors) of the measurements. The authors might provide here the plot of sera from mice immunized with known S1 antigen as reference and for the sake of a quantitative comparison.

Moreover, a co-immunoprecipitation assay might be performed to integrate the data shown in Fig. 6C and prove the binding of antibody B3-1 to the HexaPro-SS-2W.

As the same authors acknowledge in the Discussion section, more variants should be evaluated to prove that HexaPro-SS-2W might be efficacious to overcome resistance of S1-based vaccines.

Reviewer #3 (Remarks to the Author):

In this manuscript, Nuqui and colleagues combined computational, structural, and biochemical approaches to rationally design a Spike construct that resides in a stabilized, closed prefusion conformation. This is a beautiful piece of work, in which all the experiments were designed and carried out at a very high level. I have nothing to criticize and fully support publication on Nature Communications.

Reviewer #4 (Remarks to the Author):

Towards the design of a SARS-CoV-2 immunogen that is less prone to antigenic drift, Nuqui and colleagues have focused on designing a novel S2 only antigen. While the S2 immunogen can elicit neutralizing antibodies, its open conformation also tends to elicit non-neutralizers – which can be a problem when used as a vaccine construct. Here, the authors use a simulation-driven approach to design novel S2 immunogens in the closed and pre-fusion conformations, which they hypothesize can provide protection by eliciting neutralizing antibodies. To test this, the authors prime and boosted immunized mice with their two candidate vaccines, where one group was immunized with the unmodified S2 (HexaPro-SS- Δ stalk) and the other with the modified S2 that contains two double WW mutations on T998 and V991 (HexaPro-SS- Δ stalk 2W).

The overall study is quite elegant, well explained and in vivo experiments were straight forward.

However, there are parts of the in vivo experiments that should be addressed. This can provide a better assessment of the feasibility of having an S2 only antigen and the impact of the 2W mutation on breadth, neutralization and protection.

Given the hypothesis that S2 can elicit a broader breadth of antibody response, it would benefit the community if the authors can demonstrate that immunization by either constructs have comparable binding to full-length spike proteins between the different VoCs or other betacoronaviruses. It would also have been ideal to have a regular HexaPro-immunized group to compare the immunogenicity of these new candidate vaccines with that of a full-length version of the construct. It may also provide some clues whether the elicited response can lead to protection against challenge.

With regards to the way the neutralization titers are presented – it would be more beneficial if they are presented as half maximal inhibition values. Using AUC is not the usual convention in presenting neutralization titers. The shape of a curve is arguably more important than the area for assessing the neutralization. Given that viral infection was quantified by counting GFP positive cells using a plate reader, that data can be easily converted to a percent inhibition (compared to the no antibody control) followed by the calculation of the half maximal inhibition value. This method will allow the authors to more appropriately and easily compare their data with other studies.

Response to the reviewers

For reference, the comment responses are shown in blue and specific revisions performed to the manuscript or supplementary information are highlighted in red.

Reviewer: 1

This is a very interesting, up to date, and exciting paper on the redesign of stabilized S2 variants of the Sars-CoV-2 spike protein. The Authors combine long classical MD simulations with WE analysis to profile the opening mechanism of the protein.

The data are used to infer mutations able to fill voids in S2, with the working hypothesis of stabilizing this domain.

The authors validate their design with Cryo-EM, thermostability and expression data. Importantly, stabilized variants show enhanced immunogenicity also against distinct variants.

I found this paper very nice and interesting to read and I find it suitable for Nat Comm. The authors might consider expanding even more the generality of their discussion and implications by discussing their data in the context of integrating methods to discover new epitopes against distinct variants, or to integrate other types MD-based analyses to characterize stability.

Response: We thank the Reviewer for the positive feedback on our work. We also appreciate and value the comment about expanding the discussion of the implications of these data. The revised version of the manuscript now expands the discussion to contextualize our work among additional computational approaches used to study emergent VOCs.

Reviewer: 2

In this article, Nuqui et al. present the design of a variant of the subunit S2 of the SARS-CoV-2 spike protein with great potential as immunogen for vaccine development that could address the inefficacy towards the emerging virus variants of vaccines based on the less conserved subunit S1. The design is carefully described and originated by a thorough computational investigation using a number of advanced calculations. The in silico results are supported by experiments, including the resolution of a Cryo-EM structure and integrated by both in vivo and in vivo assays. The work is technically solid, clearly written, and the conclusions supported by the results. These aspects together with the relevance of the discovery make the manuscript of great interest and I suggest its publication in Nature Communications.

Response: We thank the Reviewer for highlighting the strengths of our work and for the positive feedback.

A number of points to improve the quality of the work are reported below.

- A cartoon representation of the functional motion of the Spike protein, reported as Fig. 1, would be beneficial for the reader;

Response: We appreciate the suggestion of the Reviewer about including a panel with a cartoon representation of the functional motion of the Spike protein in Fig. 1. However, we are concerned that representing the functional motions of the spike protein in Fig. 1 would be misleading as the full spike was not studied here and would highlight the S1 subunit, that is not the focus of the study. What we studied was the opening of the S2 subunit; Movie S1 shows the trimer opening motion of interest and Fig. 2 illustrates the major conformations along the opening transition. Fig. 1 is intended to show the S2 system in the context of the spike and to highlight the engineered sequence modifications present in the initial (base) construct, which is further stabilized in the present study.

- Page 6, line 24 - the fusion peptide's residues should be highlighted in the original Fig. 1;

Response: We appreciate the suggestion of the Reviewer to highlight the fusion peptide in Fig.1. We kindly disagree with the modification as it would be out of context. We specifically highlight the mutations leading to an engineered S2, which do not include any modifications to the fusion peptide. Page 6, Line 24 simply describes how the model for simulation was built, and we did not investigate the role of the fusion peptide here.

- Figure 3 - It would be interesting to show the contact values as function of the simulation time. This analysis might provide insight into the opening mechanism of S2.

In addition, the partially open and the open conformation of S2 should be shown in this figure together with the closed one in order to appreciate the structural differences;

Response: We appreciate the suggestion of the Reviewer to show the contact values as a function of simulation time. We agree that highlighting the time-dependent changes could provide important context for the opening mechanism and we initially thought to show time evolutions; however, the simulation trajectories of the WE method cannot be appropriately represented in this format. For the heatmaps in Fig. 3, we analyzed ~200 opening pathways, and they each have different opening durations ranging from ~90 ns to ~312 ns. A time evolution plot would show 30 contact score traces (10 residues with 3 chain-specific contact scores) for each pathway, which would make it difficult to discern residue-specific contributions and changes upon introducing substitutions. A plot for a single system with ~6,000 traces would obfuscate key insights, and manually selecting a handful of pathways to reduce the number of traces could lead to a subjective representation. Furthermore, as these pathways also differ in the simulation time spent in the major conformations shown in Fig. 2c-e, a time-evolution representation would decouple the connection between Fig. 2 and Fig. 3 that provides crucial mechanistic details. The average distance plotted in Fig.

2 (shading of the hexagons) shows the opening, while the heatmap in Fig. 3 shows that it is asymmetric with certain residues forming weaker contacts. We selected the heatmap representation as we are already dividing the conformational space from closed to open in 3 regions (closed, partially open, open) where the transition between regions proceeds sequentially and unidirectionally. Therefore, a heatmap representation broadly accounts for the time evolution independent from the diverse opening pathways.

Regarding adding partially closed and open conformations to Fig. 3, we opted to not include these in Fig. 3 as these conformations are already shown for the base construct in Fig. 2 from a side and top perspective. The mechanism is conserved, so the major conformations indicated would be the same for the engineered systems. Showing them in Fig 3 would be repetitive and may create confusion because we show the heatmaps for the four simulated systems, so it would be easy to mislead the reader when associating molecular representations to heatmaps and vice versa. Movie S1 illustrates the transition between the conformations shown in Fig. 2 for the conclusions to be more accessible.

- *Why the transition from the closed to the open states of S2 is barrierless in the probability plots shown in Fig. S2A, S7A, S8A and S9A. Especially in the HexaPro-SS-2W, considering the thermodynamic stability of the closed state in this system, I would expect a relatively large barrier separating the two states. That would also explain the slower rate computed for the opening motion in such a system with respect to the other systems. Could the authors explain this unexpected result?*

Response: We thank the Reviewer for offering us the opportunity to clarify this important point. We would like to start with a very important consideration: as the Reviewer expects, for all the simulated systems, the transition from the closed state to the open state is *not* barrierless. Although this is already illustrated in both Figs. S2A, S7A, S8A and S9A (as mentioned by the Reviewer, corresponding to Figs. S2A, S8A, S9A, S10A in the revised version), and in Figs. S2B, S7B, S8B and S9B (corresponding to Figs. S2B, S8B, S9B, S10B in the revised version), we recognize that the plots may not be sufficiently effective to convey this point.

In detail, in Figs. S2A, S7A, S8A and S9A (corresponding to Figs. S2A, S8A, S9A, S10A in the revised version), the probability distribution of the two progress coordinates have been calculated, as an *average*, over three separate ranges of WE iterations for each system: the iterations *preceding reweighting* (top row), the iterations *in between 1st and 2nd reweighting* (middle row), and the iterations *following the 2nd reweighting* (bottom row). Looking at the top row, the energy barrier separating the closed and the open states is evident and unrealistically large (i.e., $\sim 100 k_B T$). We note that in the WE iterations considered in the first row of plots we are far from the equilibrium, which explains the large energy barrier. However, this is already an indication that the transition between the two states is *not* barrierless. Upon reweighting twice, convergence towards steady-state conditions is enhanced, and as a consequence, the energy barrier between the closed and the open state drops to values that are more plausible (i.e., $< 20 k_B T$). Unfortunately, we recognize that the energy barrier for the closed to open transition cannot be fully appreciated in the middle and bottom rows because the color bar is scaled proportionally between $k_B T$ values ranging from 0 to 300. In brief, everything in the region between the closed and the open state may look barrierless because of the way the color bar is scaled. We apologize for this inconvenience, and we agree that the plots in the middle and bottom rows may be misleading in this sense: one might not be able to appreciate the energy barrier separating the two investigated states obtained upon reweighting, i.e., when near steady-state conditions are enhanced.

In the revised version of the supplementary information, we have now introduced a new figure, Fig. S11, where the energy landscape between the closed state (marked with a “C”) and the open state (marked with an “O”) can be more easily appreciated (figure is included below for convenience). Similar to S2A, S7A, S8A and S9A (corresponding to Figs. S2A, S8A, S9A, S10A in the revised version), this figure presents, for all systems, the 2-D probability distributions of the two WE progress coordinates averaged across all iterations following reweighting. To improve clarity, we have cropped the plot to focus solely on the region of interest within the explored conformational space. Moreover, the color palette of the color bar has been scaled from 0 (minimum) to 18 (maximum) $k_B T^{-1}$. This adjustment aims to facilitate a better understanding of the roughness of the energy landscape, which was not possible in S2A, S7A, S8A, S9A (corresponding to Figs. S2A, S8A, S9A, S10A in the revised version). As one can better appreciate from this new figure, in all system, the Closed and the Open states are indeed separated by an energy barrier, which is slightly higher in the case of Hexapro-SS-T998W. As stated in the manuscript (line 14-19, page 12), T998W is the system with the slowest closed-to-open transition, closely followed by 2W and WT, whereas V991W

exhibits the fastest transition. Therefore, with regard to this aspect, T998W is the most kinetically stabilizing mutation. We note that the energy profiles for the investigated states corresponds to the trends of the calculated kinetic rates, which, as already stated in the manuscript (line 20-23, page 12), are only “estimates” that we warned are always affected to a certain extent by approximations and sampling limitations. However, these rates are consistent with each other and with other similar motions taking place in class I fusion proteins.

Furthermore, it is important to note that, although one may expect a remarkably higher free energy barrier for the closed-to-open transition in 2W or T998W due to the superior stability of their folded closed state, this correlation does not always hold true. The energy landscape of proteins is inherently intricate, and a straightforward one-to-one correlation cannot always be assumed. It is crucial to recognize that the overall stability and dynamics of a protein are shaped by numerous factors, and mutations in specific regions, such as residues at positions 991 and 998, can exert intricate and multifaceted effects on the energy landscape and transition states. Even though the closed state of the folded Hexapro-SS-2W variant exhibit greater stability than the base construct, it does not necessarily dictate the energy of its transition state. The stability of the folded closed state and the energy of the transition state are indeed distinct aspects. The energy of the transition state could vary, being either lower, higher, or remain similar. Specifically, the introduction of tryptophan residues at positions 991 and 998 of the S2 apex is stabilizing for the S2 in the closed state. However, it introduces different interactions and dynamics during the transition—should the transition occur—which modify the energy landscape accordingly. This is also the reason why V991W appears to open slightly faster, or akin to the base construct, and why the energy landscape of 2W resembles something between those of V991W and T998W. Discussions on this important aspect can be found on page 19, line 25-29 of the revised manuscript.

*- Thermodynamic and kinetic estimates obtained from WE simulations rely on the convergence of the calculations. If an investigator badly chooses the convergence point of the simulations, the obtained results might be incorrect. Considering the relevance of this point, the authors should include at least in the **Methods** section, and in the SI, how they evaluate the convergence of the WE simulations.*

Moreover, the authors wrote in different parts of the manuscript and SI “To facilitate the convergence of WE simulations toward equilibrium, we applied the WE Equilibrium Dynamics (WEED) protocol twice to reweight the trajectories”. What does this exactly mean? Reweighting is used to recompute the unbiased Boltzmann distribution (probability function), but if the calculations is not converged, even the reweighted probability is not converged. A possible explanation is that the probability plots reported in Fig. S2B and (less in S8B and S9B) show similar profiles computed at different simulation times, suggesting an approach to the convergence of the calculations. The authors should however clarify this point;

Response: We agree with the Reviewer about the importance of properly discussing how the convergence of simulations was assessed, and we apologize if this point did not clearly come across in the manuscript. We thank

the Reviewer for encouraging us to address their concerns, as this offers us the opportunity to improve the manuscript's quality.

We first would like to clarify the Reviewer's doubts about the reweighting procedure adopted in WE simulations, which was performed with the "WEED" tool implemented within WESTPA 2.0. As already thoroughly explained in *section 1.3* of the SI, WE simulations are started from a set of initial states, each one of them is assigned the same initial weight (or probability) so that the total probability is equal to 1. In our case, as described in the WE simulation protocol, we selected 50 initial states, each one carrying a 0.02 initial weight (total $P = 1$). Starting from each of these states, independent MD simulations are conducted in parallel for 100 ps. At the end of the simulations, trajectories that move to empty bins along the progress coordinates are replicated, and their weights are evenly distributed among the resulting offspring trajectories. Trajectories that fail to make progress are occasionally terminated, and their weights are combined with other trajectories that will be continued. This cycle is then repeated for many iterations. One limitation that arises when calculating kinetic quantities from WE simulations is that most of the weight (probability) will be retained by the trajectories which populate the bins corresponding to the starting state (i.e., the Closed state), whereas the few successful trajectories (i.e., the ones entering the Open state) will initially carry very small weights (10E-50, for example), which may slowly increase as more WE iterations are run and more "successful" trajectories are collected. Unfortunately, it would take an incredibly large number of iterations to reach steady-state conditions due to the long "relaxation" time required from the initial conditions. In these cases, this relaxation time can be reduced if the initial conditions are chosen to approximate the steady-state probability distribution. This is what Zuckerman and collaborators proposed in Bhatt et al., *J. Chem. Phys.* 2010 133(1) (DOI: 10.1063/1.3456985), where they outlined a "reweighting" procedure to enhance the attainment of steady-state conditions in both *equilibrium steady-state WE simulations* (like the ones performed here) and *steady-state WE simulations*. This procedure utilizes the trajectories that are generated during the ordinary WE simulations to determine the conditional probabilities (k_{ij}) to hop among bins, i.e., the inter-bin rates k_{ij} . Then, these bin-to-bin transition probabilities are used to estimate the new bins' weights to infer steady-state behavior. This "reweighting" procedure is implemented in the "WEED" and "WESS" tools within WESTPA 2.0 software. As mentioned in the manuscript and in the SI, in our WE simulations we applied this procedure twice to enhance convergence towards equilibrium steady-state conditions. Ideally, this procedure should be repeated several times. However, a crucial consideration is that one should extend the WE simulation for numerous iterations before applying the reweighting procedure again, thereby significantly escalating the cost of the simulations. It is important to note that in our case, given the substantial system size (~350,000 atoms), the number of systems (4), and the inherent complexity of the investigated process, repeating the reweighting procedure more than two times would have demanded an extraordinarily large extra amount of resources. Therefore, upon assessing the convergence of the WE simulations (see next paragraph), we decided not to extend the simulations. **In the revised version of the SI, at the end of the subsection "WE simulations of HexaPro-SS-Astalk construct," we have introduced a few sentences to better contextualize the reweighting procedure performed in the WE simulations.**

The reweighting procedure described above allows us to address the very important comment raised by the Reviewer about the convergence of our WE simulations. Convergence of WE simulations was assessed in two ways. The first one, as also correctly noticed by the Reviewer, is through the 1D probability plots reported in Figs. S2b, S7B, S8B, S9B (Figs. S2B, S8B, S9B, S10B in the revised version). For each progress coordinate, we calculated the probability distribution at three different times. Notably, a significant shift is observed between the probabilities calculated over the WE iterations preceding reweighting and those computed during the WE iterations spanning the period between the 1st and 2nd reweighting. Subsequently, after the 2nd reweighting, the probabilities tend to stabilize, showing only minor deviations from the previous values in the region of interest. This trend underscores reasonable convergence. Otherwise, we would have observed another large shift in the probabilities. The second way we assessed convergence is illustrated in the plots reported in Fig. S13 (Fig. S15 in the revised version), where the evolution of the rolling average of the kinetic rate estimated for the opening transition is presented for each system. In a scenario where the simulations are converged, the curve indicating the evolution of kinetic rate rolling average values over the WE iterations should plateau around a specific value. As it can be evinced from Fig. S13 (Fig. S15 in the revised version), especially from panel C, where the rolling averages were calculated over larger windows of iterations (window width corresponding to ~10% of the total number of WE iterations instead of one iteration only), for all systems the curve start plateauing, especially over the iterations following the 2nd reweighting. We concur that, amongst the four systems, the convergence for Hexapro-SS-2W has room for improvement. However, considering the complexity of these

simulations and the very good relative agreement amongst the four systems, the observed trends are very reassuring in this sense. Finally, we remark that a full characterization of opening kinetic rates goes beyond the purpose of this work, which was engineering an S2 trimer stabilized in the prefusion closed conformation. In the manuscript we warn the readers that, although plausible and in general agreement with each other and other similar motions, the reported kinetic rates are only estimates that could be affected, to a certain extent, by sampling limitation (line 20-23, page 12).

Lastly, convergence of free energy calculations was assessed by inspecting the intersection between the work values calculated from the forward and reverse non-equilibrium simulations. As it can be evinced from Fig. S17 of the revised version (previously Fig. S15), the forward and reverse work values sufficiently intersect in all the simulations performed, allowing accurate calculation of $\Delta\Delta G$.

As suggested by the Reviewer we have now added a few sentences about the convergence of the WE simulations in the Methods section of the revised version of the manuscript and an entire paragraph in the SI (subsection “*WE simulation summary*”). We have also added a sentence about the convergence of the free energy calculations in the respective Methods section of the revised version of the manuscript.

- Page 24, line 13 - “For the unfolded state, we used a capped tripeptide (G-X-G) built with Chimera, where X is the amino acid of interest (either V, T, or W). I do not understand why the authors simulated the tripeptide G-X-G as unfolded state to compute the free-energy difference between the folded and unfolded state (alchemical free energy calculations). That tripeptide is not representative of the real unfolded state of S2 and the estimates might be biased by this choice. A natural choice would have been to use the open state as reference to compute the free-energy difference with respect to the closed state (not folded state). This calculation would further provide an orthogonal validation of the WE calculations. The authors should motivate in the main text the reason why they opted for the tripeptide G-X-G;

Response: We thank the Reviewer for inviting us to improve the description of the protocol we followed for our alchemical non-equilibrium free energy calculations.

It is crucial to clarify that our free energy calculations aim to estimate the change in the protein’s folding free energy due to a mutation, with the protein being the S2 trimer in the closed state. Instead, the free energy associated with the closed-to-open transition was *not* investigated by means of free energy calculations. Therefore, the two legs of the thermodynamic cycle illustrated in Fig. S16 of the revised version (previously S14) should represent the *folded* and *unfolded* states, not the *closed* and the *open* states. On the other hand, while simulating the actual unfolded protein would be ideal, practical challenges hinder this approach. Firstly, the structure of the unfolded state is unknown but certainly does not coincide with the open state. Secondly, the force fields employed in MD simulations are specifically tailored for proteins in their folded state. Therefore, straying from the folded state would remarkably impact the accuracy of such calculations. A few relevant publications in this field discussing the force field limitations when simulating folding/unfolding event are available: (i) Piana et al. *Biophys J.* 2011, doi: 10.1016/j.bpj.2011.03.051; (ii) Freddolino et al. *Biophys J.* 2009 doi: 10.1016/j.bpj.2009.02.033; (iii) Fischer et al. *J. Chem. Theory Comput.* 2024, doi: 10.1021/acs.jctc.3c011060000. Using the G-X-G tripeptide is a sensible approach, as it minimizes the errors associated with simulating the unfolded state (should the unfolded state be known). We note that this approach has been commonly adopted in alchemical non-equilibrium free energy calculations and has proven to align well with experimental data in numerous studies, including the one at hand. (i) Seeliger et al. *Biophys J.* 2010 doi: 10.1016/j.bpj.2010.01.051; (ii) Gapsys and De Groot, *J. Chem. Inf. Model.* 2017, doi: 10.1021/acs.jcim.6b00498; (iii) Jones et al. *ACS Med. Chem. Lett.* 2023, doi: 10.1021/acsmchemlett.2c00517;

As encouraged by the Reviewer, we have now introduced a few sentences (and accompanying references) aimed at clarifying this point in the Methods section (“*Alchemical non-equilibrium free energy calculations*” section) of the revised version of the manuscript.

- The resolution of the Cryo-EM structure should be reported. Did the authors try to resolve at Cryo-EM the structure of the HexaPro-SS- Δ stalk S2 (wild type)? It would be interesting a comparison;

Response: We appreciate the suggestion of the Reviewer to report the resolution of the Cryo-EM structure (2.8 Å). Although the resolution was provided in Table S4, it was not shown in the main text. **We have included this information in the revised version of the manuscript (Page 16, Lines 7-8 and Page 17, Line 8).**

A separate work reporting an in-depth structural characterization of the open crystal structure of HexaPro-SS-Δstalk S2 construct was still under review at the time of submission of this manuscript. This work was conducted by part of our team and it is now published as Hsieh, CL. *et al.*, Prefusion-stabilized SARS-CoV-2 S2-only antigen provides protection against SARS-CoV-2 challenge. *Nat. Commun.* **15**, 1553 (2024) (doi: 10.1038/s41467-024-45404-x). We note that the open structure of HexaPro-SS-Δstalk S2 (wildtype) is already shown in Fig 2a of our manuscript (gray cartoons). **However, as requested by the Reviewer we have now added a new supplementary figure (Supplementary Fig. 3) in the revised version of the SI where we overlay the open conformation obtained from the WE simulation of HexaPro-SS-Δstalk S2 with the crystal structure of the open HexaPro-SS-Δstalk S2 solved by Hsieh et al. (PDB ID: 8U1G) (also included below for convenience).**

- Page 17, line 24 - This part is not clear in my opinion. The decreased infectivity seems to be not so strong looking at the histograms in Figure 6B, also considering the uncertainty (errors) of the measurements. The authors might provide here the plot of sera from mice immunized with known S1 antigen as reference and for the sake of a quantitative comparison.

Moreover, a co-immunoprecipitation assay might be performed to integrate the data shown in Fig. 6C and prove the binding of antibody B3-1 to the HexaPro-SS-2W.

As the same authors acknowledge in the Discussion section, more variants should be evaluated to prove that HexaPro-SS-2W might be efficacious to overcome resistance of S1-based vaccines.

Response: We appreciate the suggestion to include a comparison of Hexapro-SS-2W against S1-based antigens across additional variants. We agree that these additional studies will be important to investigate the immunogenicity of Hexapro-SS-2W for consideration as a vaccine candidate for further development. Although these data would illustrate the antigenicity of Hexapro-SS-2W in the broader context of vaccine development, we believe it is beyond the scope of this work as the goal was to close the S2 apex rather than enhance its immunogenicity. In our work we sought to compare the immunogenicity of open S2 (HexaPro-SS) against a closed S2 (HexaPro-SS-2W). Fig. 6b clearly shows that the immunogenicity of HexaPro-SS is not impacted by introduction of the 2W substitutions. We note that despite the neutralization strength, the decreased infectivity from HexaPro-SS-2W immunization produces statistically significant neutralization relative to the control. The statistical tests are described in the figure caption. Furthermore, a recent work published by part of our team (Hsieh, CL et al., *Nature Communications* **15**, 1553 (2024), doi: <https://doi.org/10.1038/s41467-024-45404-x>), fully addresses the point raised by the Reviewer as it compares the immunogenicity of an open S2 against a spike construct that also includes S1 (HexaPro); the paper reports extensive

immunization and mouse protection with different prime-boost strategies between HexaPro (S1 and S2), HexaPro-S2 (S2-only), and HexaPro-SS (S2-only with stabilizing disulfides).

Regarding co-immunoprecipitation of antibody B3-1 to Hexapro-SS-2W, a manuscript including a cryo-EM structure of B3-1, BLI data of B3-1 binding to various SARS-CoV-2 spike constructs (S-2P, HexaPro, HexaPro 3-RBD locked down), and SPR data of B3-1 with HexaPro-SS is in preparation by some authors of our team. Instead, the BLI experiments performed in the present work and displayed in Fig. 6c indicate that binding of B3-1 is reduced in HexaPro-SS-2W when compared to HexaPro-SS. As hypothesized in the manuscript, this is most likely a consequence of HexPro-SS-2W adopting a closed conformation (Lines 15-17, Page 18). Additional characterization of the binding mode of B3-1 to Hexapro-SS-2W would exceed the scope of this work.

Reviewer 3:

In this manuscript, Nuqui and colleagues combined computational, structural, and biochemical approaches to rationally design a Spike construct that resides in a stabilized, closed prefusion conformation. This is a beautiful piece of work, in which all the experiments were designed and carried out at a very high level. I have nothing to criticize and fully support publication on Nature Communications.

Response: We thank the Reviewer for the positive feedback on our work.

Reviewer 4:

Towards the design of a SARS-CoV-2 immunogen that is less prone to antigenic drift, Nuqui and colleagues have focused on designing a novel S2 only antigen. While the S2 immunogen can elicit neutralizing antibodies, its open conformation also tends to elicit non-neutralizers – which can be a problem when used as a vaccine construct. Here, the authors use a simulation-driven approach to design novel S2 immunogens in the closed and pre-fusion conformations, which they hypothesize can provide protection by eliciting neutralizing antibodies. To test this, the authors prime and boosted immunized mice with their two candidate vaccines, where one group was immunized with the unmodified S2 (HexaPro-SS- Δ stalk) and the other with the modified S2 that contains two double WW mutations on T998 and V991 (HexaPro-SS- Δ stalk 2W).

The overall study is quite elegant, well explained and in vivo experiments were straight forward. However, there are parts of the in vivo experiments that should be addressed. This can provide a better assessment of the feasibility of having an S2 only antigen and the impact of the 2W mutation on breadth, neutralization and protection.

Response: We thank the Reviewer for the positive feedback on our work.

Given the hypothesis that S2 can elicit a broader breadth of antibody response, it would benefit the community if the authors can demonstrate that immunization by either constructs have comparable binding to full-length spike proteins between the different VoCs or other betacoronaviruses. It would also have been ideal to have a regular HexaPro-immunized group to compare the immunogenicity of these new candidate vaccines with that of a full-length version of the construct. It may also provide some clues whether the elicited response can lead to protection against challenge.

Response: We appreciate the Reviewer's suggestion to show serum binding to the full-length spike proteins of a larger panel of SARS-CoV-2 VOCs and other betacoronaviruses. However, we feel that the data presented in the manuscript (neutralization assays with rVSV-SARS-CoV-2 Wuhan and Omicron BA.1 in Fig. 6b and Supplementary Fig. 24 in the revised SI) fulfill our key objective, i.e., to demonstrate that the engineered S2 variants described herein do not negatively impact the S2-directed antibody response. As remarked on lines 15-19, page 20, given that this work is just a starting point towards the development of optimized S2 antigens with broad protective potential, we believe that a fuller examination of antiviral breadth with the current antigens is beyond the scope of the current manuscript.

With regards to the way the neutralization titers are presented – it would be more beneficial if they are presented as half maximal inhibition values. Using AUC is not the usual convention in presenting neutralization titers. The shape of a curve is arguably more important than the area for assessing the neutralization. Given that viral infection was quantified by counting GFP positive cells using a plate reader, that data can be easily converted to a percent inhibition (compared to the no antibody control) followed by the calculation of the half maximal inhibition value. This method will allow the authors to more appropriately and easily compare their data with other studies.

Response: We concur with the Reviewer that neutralization IC50s are often presented but would also point out that there is no standard way to represent neutralizing activity, particularly with polyclonal antisera. Indeed, AUCs have been frequently used for this purpose by others and us (Yu et al., *Stat. Biopharm. Res.* 2012, doi: 10.1080/19466315.2011.633860; Lasso et al., *PLoS Comput. Biol.* 2022, doi: 10.1371/journal.pcbi.1009778; Schommers et al., *Nat. Med.* 2023, doi: 10.1038/s41591-023-02582-3; Dogan et al., *Commun. Biol.* 2021, doi: 10.1038/s42003-021-01649-6). Furthermore, when examining statistical approaches to analyze HIV-1 neutralizing antibody assay data, Yu and co-authors concluded that ‘*the AUC method is more powerful than the IC50 method.*’ They further noted that ‘*Unlike IC50, [...] AUC summarizes the neutralization responses across the entire concentration range without requiring assumptions about the shape of the titration curve. In contrast, IC50 measures the neutralization activity at a single point and is easily interpretable only when titration curves are sigmoidal shaped within the concentration range, which are often not the case.*’

In this instance, we elected to use AUCs precisely because the shape of the curves arising from the generally low levels of neutralization observed with S2-directed immune sera makes them less suitable to logistic curve fits—IC50s calculated from such curve fits displayed large 95% confidence intervals.

We agree with the Reviewer that the shape of the curve can be important. Accordingly, we have included the raw neutralization curves in Supplementary Fig. 24 of the revised Supporting Information so that the reader may directly assess their shapes and other properties.

REVIEWER COMMENTS

Reviewer #2 (Remarks to the Author):

Regarding the first three points raised by the Reviewer, the authors did not modify Figures 1 and 3 as requested to make those figures and the text clearer. According to them these modifications are not necessary. I am still convinced they are.

Regarding the calculation of the contacts as a function of the time, they preferred not to do that. I do not understand the provided explanation, however I think it is useful to report such data that could provide insight into the mechanism of conformational changes of S2.

I also asked to clarify the presence of barriers between the closed and open states identified in the different systems. The authors provided an additional figure highlighting the difference in energy between states. The energy profiles do not demonstrate the stabilization of the closed state in the Trp variants, especially in 2W and V991W, with respect to the basal construct, which is the claim of the work. From these plots, the energy differences between closed and open states computed in the basal and the mutated constructs are similar, and their difference is likely lower than the method error. The reported figures provide thermodynamic (not kinetic) information and "the stability of the folded closed state and the energy of the transition state are indeed distinct aspects" as stated by the same authors. Also looking at Fig. S15, the close-to-open kinetic rates of the basal and mutant constructs are very close. No statistically significant difference might be derived from these data. The concrete risk is that the simulations do not support the research hypothesis.

Regarding the convergence of the calculations, the authors acknowledge that a rigorous assessment of the convergence of the simulations through reweighting procedure of the simulation trajectories cannot be fully accomplished due to the size of the system. I see the point, however a zoom on the 1D energy profiles after the first and second reweighting between 0 and 20 kbT highlighting the energy difference between the closed and open state, should be provided. This would provide at least a qualitative comparison of the energetic profiles.

Please provide in the caption of Fig. S3 the RMSD value between the simulated and experimental open state structure of HexaPro-SS- Δ stalk.

Looking at the plots of the work, the forward and backward processes have a difference in average energy of 20 kJ/mol at least. In a reversible process, as supposed to be the forth and back alchemical transformation of S2, the two distributions should overlap (not only the tail).

Experiments on sera from mice immunized with known S1 antigen for comparison (similar request was done by the other Reviewer) and co-immunoprecipitation assay suggested to prove the binding of antibody B3-1 to the HexaPro-SS-2W were not performed.

Regarding the IC50 values asked by the other Reviewer, the curves shown in the new Fig. S24 does not clearly indicate the desired effects of the engineered S2 trimers. All the curves are almost overlapping, including the control PBS.

Reviewer #4 (Remarks to the Author):

The authors have addressed this reviewer's concern and fully support the publication of the study in Nature Communications.

REVIEWER COMMENTS

For reference, the comment responses are shown in blue and specific revisions performed to the manuscript or supplementary information are highlighted in red.

Reviewer #2 (Remarks to the Author):

Regarding the first three points raised by the Reviewer, the authors did not modify Figures 1 and 3 as requested to make those figures and the text clearer. According to them these modifications are not necessary. I am still convinced they are.

Response: We kindly disagree with the Reviewer. In this study, we are not investigating the spikes' functional motions, nor are we investigating the dynamics of the fusion peptide. Therefore, we are not inclined to modify Figure 1 to highlight these aspects, which we consider as a distraction to the focus of our study. Furthermore, the partially open and open conformations of S2 are already clearly depicted from two distinct viewpoints in Figure 2; thus, we are not inclined to add a repetition of these representations in Figure 3, as also previously suggested.

Regarding the calculation of the contacts as a function of the time, they preferred not to do that. I do not understand the provided explanation, however I think it is useful to report such data that could provide insight into the mechanism of conformational changes of S2.

Response: We appreciate the reviewer acknowledging that they do not understand our provided explanation. We wish to share two references with the reviewer to aid in their understanding of our work and to better contextualize our analysis and methods within the field of molecular simulation. The first reference we wish to share is a comprehensive review of the theoretical basis for weighted ensemble simulations, which we use extensively here to elucidate the detailed dynamics of S2 in various states: Zuckerman and Chong, Weighted Ensemble Simulation: Review of Methodology, Applications, and Software, *Ann Reviews of Biophys*, Vol. 46:43-57, 2017. The second reference is a well-written tutorial that elaborates on the practical application of the weighted ensemble method and related analyses: Bogetti et al., A Suite of Tutorials for the WESTPA 2.0 Rare-Events Sampling Software [Article v2.0]. *Living Journal of Computational Molecular Science*, 5(1), 1655. <https://doi.org/10.33011/livecoms.5.1.1655>, 2023. We are confident that a study of these papers will alleviate the reviewer's concerns about our analysis and responses to their requests.

In this context, as we discussed in detail in our first response, the weighted ensemble simulations generate many hundreds of opening pathways (in this case 711 opening pathways are sampled across four systems). Thus, the reviewer's request to represent the time evolution of the contacts established by the 10 residues of interest currently highlighted in Figure 3 for all 711 opening pathways obtained across the four WE simulations (180 + 149 + 187 + 195) would not

result in any humanly interpretable results. It would imply depicting precisely 7110 (seven thousand-one hundred-ten) lines in a single plot (10 residues x 711 opening pathways), or ~1800 lines for each system. Instead, to fully appreciate the ensemble nature of the opening pathways, we opted for a *comprehensive overview of these contacts* that is provided by the heatmaps currently reported in Figure 3. The heatmaps show the contact score for the relevant residues in each state (closed/partiallyOpen/Open) in each system, accounting for the statistics of all the opening pathways and providing clear mechanistic information. Furthermore, we note that analyzing the ensemble of generated pathways is the field standard for the weighted ensemble method.

Moreover, the length of the opening trajectories varies (from ~90 ns to ~312 ns) and does not directly correspond to the same state across all the trajectories. For example, a “short” opening pathways occurs in 100 ns and a “long” one in 200 or 300 ns. At the 100 ns time point, the former would show zero contacts (as it is now in the open state) whereas the latter would show nonzero contacts (as it is in the closed or partially open state) resulting in a confusing depiction of the mechanism. A time course representation to describe the mechanism would be unclear because the state the system is sampling is not the same at the given time point. This is a direct consequence of the weighted ensemble method that we are using to explore the dynamics. Again we refer the reviewer to the two references provided in the first paragraph under this request.

For these reasons we are not inclined to modify Figure 3.

I also asked to clarify the presence of barriers between the closed and open states identified in the different systems. The authors provided an additional figure highlighting the difference in energy between states. The energy profiles do not demonstrate the stabilization of the closed state in the Trp variants, especially in 2W and V991W, with respect to the basal construct, which is the claim of the work. From these plots, the energy differences between closed and open states computed in the basal and the mutated constructs are similar, and their difference is likely lower than the method error. The reported figures provide thermodynamic (not kinetic) information and “the stability of the folded closed state and the energy of the transition state are indeed distinct aspects” as stated by the same authors. Also looking at Fig. S15, the close-to-open kinetic rates of the basal and mutant constructs are very close. No statistically significant difference might be derived from these data. The concrete risk is that the simulations do not support the research hypothesis.

Response: The Reviewer previously asked us if there was a free energy difference between the Closed and the Open states, and we fully addressed this point in the previous revision by including a new Figure (Figure S11).

We would like to emphasize that Figure S11 is **not** intended to show that the Closed state is stabilized in the variants, and we would like to stress that it is not appropriate to use Figure S11 to evaluate that claim. This Figure highlights that, likely due to the similar free energy barriers,

when the engineered systems transition to the open state, they do so in a similar fashion as the basal system. We emphasize that, biologically, all the systems still transition to Open since no system has been locked in the Closed state. Yet there are subtle differences. In detail, T998W opens 10-fold slower, 2W in a comparable way and V991W slightly faster (but still comparable) than the basal system. This is why, in Figure S11, the free “energy differences between closed and open states computed in the basal and the mutated constructs are similar” (to quote the Reviewer’s words), especially in the case of 2W and V991W. We note that the free energy landscapes represented in Figure S11 correlate with the rate constants of the associated closed-to-open transitions, which have already been reported (**with errors included**) in the main text, Fig. S15, and Table S3. We note that this point is already thoroughly discussed in the main text (line 14-19 page 12) and has also been addressed in the previous extensive point-by-point response. It highlights that **only T998W is kinetically stabilizing**, i.e., slows down the open-to-closed transition / increases the free energy barriers between Closed and Open states. We remark that slowing down the closed-to-open transition is distinct from the relative stability of the systems’ folded Closed states. As highlighted in the previous response, the stability of the folded Closed state of a system and the energy of the transition state separating the Close and Open states are indeed distinct aspects. Moreover, the reverse transition “open-to-close” or “closing” (not investigated in this paper because beyond the scope) could further contribute to shifting the equilibrium towards Closed for T998W and 2W.

Here is the extract from the main text where we discuss the rate constants associated to the opening transition, their limitations, and the agreement with each other and other similar motions:

“Although the rate constant does not significantly change for HexaPro-SS-V991W [$k_{\text{opening}} = (1.1 \pm 0.5) \times 10^5 \text{ s}^{-1}$], it becomes smaller for HexaPro-SS-2W [$k_{\text{opening}} = (7.0 \pm 2.3) \times 10^3 \text{ s}^{-1}$] and HexaPro-SS-T998W [$k_{\text{opening}} = (3.4 \pm 0.1) \times 10^3 \text{ s}^{-1}$], indicating that T998W induces a roughly 10-fold slower trimer opening. V991W alone does not appear to affect opening kinetics. Although we acknowledge that these estimations might be affected by sampling limitations, the calculated values align well with each other and single-molecule fluorescence resonance energy transfer (smFRET) studies reporting similar domain movements of class I fusion glycoproteins occurring in the millisecond-to-second time scales”.

Finally, what proves **the relative stability of the folded Closed state** in the tested variants with respect to the base system are the Free Energy Calculations performed on the Closed state for each system (Fig. 4A), which strikingly correlate with the experimental data and fully support the findings. Our findings calculations are supported by thermostability experiments, expression yields, and cryoEM structural data.

Regarding the convergence of the calculations, the authors acknowledge that a rigorous assessment of the convergence of the simulations through reweighting procedure of the simulation trajectories cannot be fully accomplished due to the size of the system. I see the point,

however a zoom on the 1D energy profiles after the first and second reweighting between 0 and 20 $k_B T$ highlighting the energy difference between the closed and open state, should be provided. This would provide at least a qualitative comparison of the energetic profiles.

Response: We would like to clarify that we have not stated that a rigorous assessment of the convergence of the simulations cannot be fully accomplished. In the previous, extensive response addressing this point, we instead acknowledged that convergence has always room for improvement, albeit the observed trends shown by the kinetic rate plots (Figure S15) and the 1D free energy plots (Figures S2B, S8B, S9B, S10B) are reassuring in this sense. Convergence was, in fact, carefully assessed, and following the Reviewer's previous suggestion, we have already detailed how we did that in the previous response, in the main text, and in the Supplementary Information. We would like to remark that, to ensure we could get an improved and better convergence, we have conducted the WE ensemble simulations for a staggering 185 μs aggregate sampling (see Table S3), obtaining a total of 711 independent opening pathways.

As further suggested by the Reviewer, we have now added a magnified view of the 0 to 20 $k_B T$ region of the 1D energy plots. We note that, after 2nd reweighting, the differences with the energies after 1st reweighting in the regions of interest are usually smaller than 5 $k_B T$ and, in some cases, even smaller than 2.5 $k_B T$, which is not a dramatic change. In scenarios where the simulations are far from convergence, larger energy fluctuations would be observed (see the jumps between the energies before reweighting and the energies after 1st reweighting). **The updated panels are now included in Supplementary Figure 2B, Supplementary Figure 8B, Supplementary Figure 9B, Supplementary Figure 10B. For convenience, the updated plots are also provided below.**

T998W

2W

Please provide in the caption of Fig. S3 the RMSD value between the simulated and experimental open state structure of HexaPro-SS- Δ stalk.

Response: We are happy to clarify this point. The RMSD of the C-alpha atoms of the S2 central helices (residues 987-1033) is 1.64 \AA . Instead, considering all the S2 residues, the RMSD of the C-alpha atoms is 3.76 \AA . We have added this information to the caption of Supplementary Figure S3.

Looking at the plots of the work, the forward and backward processes have a difference in average energy of 20 kJ/mol at least. In a reversible process, as supposed to be the forth and back alchemical transformation of S2, the two distributions should overlap (not only the tail).

Response: It is unclear what the Reviewer is trying to suggest by mentioning the 20 kJ/mol energy difference and we do not understand how that is relevant to the convergence of the reported free energy calculations. The histograms depicting the forward and reverse work values (Supplementary Figure 17) show a satisfactory convergence, evident in their overlapping, fulfilling the Crooks fluctuation theorem's criteria (here we refer the reader to Crooks, Entropy production

fluctuation theorem and the nonequilibrium work relation for free energy differences, *Physical Review E* 60.3: 2721 (1999)). To ensure this convergence, we conducted 1000 simulations in each direction, meaning that the observed overlap, referred to by the reviewer as a “tail,” represents the aggregation of hundreds of simulations in both directions. In fact, the histograms reported here represent an ideal outcome of such calculations. We refer the reviewer to the comprehensive chapter by Aldeghi et al., offering thorough insights into the theoretical framework and practical applications of alchemical free energy calculations in protein design: Aldeghi, M., de Groot, B.L., Gapsys, V., Accurate Calculation of Free Energy Changes upon Amino Acid Mutation. In: Sikosek, T. (eds) *Computational Methods in Protein Evolution. Methods in Molecular Biology*, vol 1851. Humana Press, New York, NY. (2019), Link: https://link.springer.com/protocol/10.1007/978-1-4939-8736-8_2 (with particular attention to the histogram in Figure 5 of this chapter, which is described as a standard outcome).

Experiments on sera from mice immunized with known S1 antigen for comparison (similar request was done by the other Reviewer) and co-immunoprecipitation assay suggested to prove the binding of antibody B3-1 to the HexaPro-SS-2W were not performed.

Response: We understand the Reviewer’s perspective on the significance of obtaining these immunogenicity data, and we agree that these experiments would be important towards the development of optimized S2 antigens with broad protective potential. We believe that a fuller examination of antiviral breadth with the current antigens is beyond the scope of the current manuscript as we do not claim that Hexapro-SS-2W is broadly neutralizing, nor has improved immunogenicity, nor speculate on differences in immunogenicity of a full-length spike (S1+S2), In line with this, the characterization of the binding mode of B3-1 to Hexapro-SS-2W would exceed the scope of this work.

Regarding the IC50 values asked by the other Reviewer, the curves shown in the new Fig. S24 does not clearly indicate the desired effects of the engineered S2 trimers. All the curves are almost overlapping, including the control PBS.

Response: We kindly disagree with the Reviewer as *the goal of the work was not to enhance immunogenicity of the S2 trimer but to stabilize the trimer in a closed state (which is supported by the simulations, thermostability experiments and the cryo-EM structure)* that does not negatively impact the S2-directed antibody response (which is supported by the BLI results). Furthermore, we do not claim that the immunogenicity is improved relative to the base construct—the results in Fig. 6b show no significant differences between the base construct and Hexapro-SS-2W. Indeed, the individual sera sample curves in Fig. S24 show overlap with the control PBS and alone do not visibly indicate differences in immunogenicity.

Although the IC50 is often reported to compare the information from these individual curves, the shapes of the curves obtained in these studies are less suitable for a logistic curve fit needed for that calculation. Therefore, the area under the curve (AUC) is calculated instead. The ordinary one-way ANOVA with Tukey's multiple comparisons test performed on the AUC indicates that Hexapro-SS-2W exhibits statistically significant differences against the PBS control, albeit weakly neutralizing, which agrees with the goal of the work.

Finally, we would like to remark that the other Reviewers fully agreed with our explanations regarding these points.

Reviewer #4 (Remarks to the Author):

The authors have addressed this reviewer's concern and fully support the publication of the study in Nature Communications.

Response: We appreciate the Reviewer's suggestions and thank the Reviewer for the feedback on our work.

REVIEWERS' COMMENTS

Reviewer #1 (Remarks to the Author):

The authors have satisfactorily addressed all the points raised by Ref 2 in their previous revision and in the new one.

Given the complexity and dimensions of the systems studied here, what the authors have accomplished in this paper represents a remarkable example of what simulations can do in combination with experiments.

Many of the things that are asked by Ref 2 concern supplementary figures with requests that often are technically apparently unsound (the contacts question for instance). In other cases (convergence, free-energies) the data have already been shown by the authors and have been based on a number of techniques and papers that are well accepted by the literature.

Finally, as an author myself, it is hard to appreciate other reviewers "wanting" their comments to be compulsorily taken up by the authors. The authors are intended to respond to the referees' concerns but should not feel obliged to modify their manuscript if they feel it is unnecessary or not strictly needed.

REVIEWER COMMENTS

For reference, the comment responses are shown in blue.

Reviewer #1 (Remarks to the Author):

The authors have satisfactorily addressed all the points raised by Ref 2 in their previous revision and in the new one.

Given the complexity and dimensions of the systems studied here, what the authors have accomplished in this paper represents a remarkable example of what simulations can do in combination with experiments. Many of the things that are asked by Ref 2 concern supplementary figures with requests that often are technically apparently unsound (the contacts question for instance). In other cases (convergence, free-energies) the data have already been shown by the authors and have been based on a number of techniques and papers that are well accepted by the literature.

Finally, as an author myself, it is hard to appreciate other reviewers "wanting" their comments to be compulsorily taken up by the authors. The authors are intended to respond to the referees' concerns but should not feel obliged to modify their manuscript if they feel it is unnecessary or not strictly needed.

Response: We thank the Reviewer for the positive feedback and for highlighting the strengths of our work.

Additional reviewer note to editor:

The referee suggests transforming data to percent inhibition by comparing the two vaccination groups and calculate how much more neutralizing they are when compared to the PBS. In the end, in Figure 6B will have two bars per virus and state how much more neutralizing it is compared to the non-immunized controls. The referee was uncertain where a 'virus-only' control was performed and this would be a suitable way to present the data without this explicit control.

However- it appears this is a matter of presentation of the results so we will leave this up to you. Where is necessary to caveat the results we would like you to do so.

Response: As the AUCs shown in Fig. 6b represent normalized data of entire neutralization curves rather than infection numbers, presenting the results as a % neutralization does not lend itself to a clear conclusion. In order to normalize the vaccine group to the PBS control, the data would need to be averaged from the PBS mice, which would not show the scatter on the PBS control and thus become less transparent. Additionally, performing a normalization to the PBS control on already normalized and transformed data would involve a second layer of normalization and transformation, complicating the interpretation of the %AUC. Consequently, the meaning of %AUC becomes ambiguous. For these reasons, we have decided not to change how data are presented in Figure 6b.